# Altered heparan sulfate metabolism during development triggers dopamine-dependent autistic-behaviours in models of lysosomal storage disorders

Maria De Risi[1,2], Michele Tufano[1], Filomena Grazia Alvino[1], Maria Grazia Ferraro[1], Giulia Torromino[1,2], Ylenia Gigante[1], Jlenia Monfregola[1], Elena Marrocco[1], Salvatore Pulcrano[3], Lea Tunisi[4], Claudia Lubrano[5], Dulce Papy-Garcia[6], Yaakov Tuchman[7], Alberto Salleo [7], Francesca Santoro [5], Gian Carlo Bellenchi[3], Luigia Cristino [4], Andrea Ballabio [1], Alessandro Fraldi[1] & Elvira De Leonibus [1,2✉]

Lysosomal storage disorders characterized by altered metabolism of heparan sulfate, including Mucopolysaccharidosis (MPS) III and MPS-II, exhibit lysosomal dysfunctions leading to neurodegeneration and dementia in children. In lysosomal storage disorders, dementia is preceded by severe and therapy-resistant autistic-like symptoms of unknown cause. Using mouse and cellular models of MPS-IIIA, we discovered that autistic-like behaviours are due to increased proliferation of mesencephalic dopamine neurons originating during embryogenesis, which is not due to lysosomal dysfunction, but to altered HS function. Hyperdopaminergia and autistic-like behaviours are corrected by the dopamine D1-like receptor antagonist SCH-23390, providing a potential alternative strategy to the D2-like antagonist haloperidol that has only minimal therapeutic effects in MPS-IIIA. These findings identify embryonic dopaminergic neurodevelopmental defects due to altered function of HS leading to autistic-like behaviours in MPS-II and MPS-IIIA and support evidence showing that altered HS-related gene function is causative of autism.

[1] Telethon Institute of Genetics and Medicine, Pozzuoli, Naples, Italy. [2] Institute of Biochemistry and Cell Biology, CNR, Monterotondo Scalo, Rome, Italy. [3] Institute of Genetics and Biophysics, CNR, Naples, Italy. [4] Institute of Biomolecular Chemistry, CNR, Pozzuoli, Naples, Italy. [5] Center for Advanced Biomaterials for Healthcare, Istituto Italiano di Tecnologia, Naples, Italy. [6] Université Paris-Est Créteil Val de Marne (UPEC), Créteil, France. [7] Department of Materials Science and Engineering, Stanford University, Stanford, CA, USA. ✉email: elvira@deleonibus.it

Mucopolysaccharidosis IIIA (MPS-IIIA) is a severe inborn metabolic error caused by mutations of the sulfamidase gene (SGSH), a lysosomal enzyme that participates in the metabolism of the glycosaminoglycan (GAG), heparan sulfate (HS)[1]. Defective sulfamidase activity leads to the accumulation of undegraded HS in the lysosome, which together with autophagy is one of the main cellular degradative systems. Lysosomal HS accumulation results in defective autophagosomal/lysosomal degradative capacities and build-up of primary and secondary aggregates (storages), ultimately leading to neurodegeneration[2] and dementia in children[1]. The vast majority of the pre-clinical literature has aimed to offer corrective therapies by targeting lysosomal dysfunction and/or substrate reduction in order to prevent neurodegeneration and dementia in MPS-IIIA.

However, dementia is preceded by severe and incapacitating autistic-like behaviours[3]. These early behavioural symptoms manifest in all forms of MPS-III (from A to D) and consist of restless, destructive, chaotic, anxious, aggressive and challenging behaviours, which are difficult to manage, have a dramatic impact on parental distress, and are resistant to behavioural therapies. Autistic behaviours are so pervasive in MPS-III that they lead to a misdiagnosis of autism spectrum disorder (ASD)[4]. They also manifest in other forms of mucopolysaccharidosis (MPS) characterized by defective metabolism of HS, such as MPS-II and MPS-I, caused by disruptions to the enzymes iduronate-2-sulfatase and α-L-iduronidase, respectively, which leads to accumulation of HS and other GAGs; for this reason, these diseases are accompanied by a faster shift from autistic- to dementia-like symptoms, meaning autistic-like symptoms can be harder to detect[5,6]. Despite this, the mechanisms leading to autistic behaviours in MPS-IIIA have not been investigated yet.

The complete lack of pre-clinical studies and therapeutic options for behavioural symptoms in MPS-IIIA has led clinicians to borrow pharmacological treatments intended for non-genetic psychiatric syndromes that show similar symptoms. Typical and atypical antipsychotics, such as haloperidol and risperidone, are dopaminergic drugs thought to exert their effects by inhibiting the action of dopamine (DA) projections, originating in the mesencephalon, on D2 receptors (D2Rs) in the prefrontal cortex and the striatum. However, they have few or variable therapeutic effects on autistic-like behaviours in MPS-IIIA, with most success against hyperactivity in very young children; however, they have been reported to have considerably higher than expected extrapyramidal side effects in older patients[7–9], which suggests disease-specific alterations in the dopaminergic systems of MPS-IIIA patients.

HS and its conjugated proteins (HSPGs) are GAGs composing the extracellular matrix that surrounds DA neurons in the mesencephalon[10], suggesting that HS might play an important role in the regulation of the DA system. Previous studies showed that HS modulates DA in PC12 cells[11,12]. More recent evidence demonstrated that the downregulation of glypican 4, one of the HSPGs, increases the differentiation of embryonic stem (ES) cells into DA neurons[13]. The mechanisms through which HS modulates DA signalling and development are as yet unknown. However, the roles of HS and HSPGs as co-receptors of many fibroblast growth factors (FGF), such as FGF2 and FGF8, which are crucial for the development of the dopaminergic system, have been already established[13–16]. Of note, HS-FGF2 signalling is impaired in cells of MPS-IIIB and MPS-I patients[17,18]. However, the impact of the impairment of HS signalling on the regulation of the dopaminergic system and, consequently, on autistic behaviours in MPS-III is currently not understood.

Understanding the disease mechanisms leading to autism in MPS-III is highly relevant in the field of ASD, since HS function is an integral part of the core synaptic organizing complexes neurexin and neuroligin, whose genes (NRXN and NLGN, respectively) are mutated in autism[19]; moreover, altered metabolism of HS has been found in both animal models and in patients with iatrogenic ASD[20,21].

Using the $Sgsh^{-/-}$ mouse model (hereafter called MPS-IIIA), which bears a spontaneous mutation in the sulfamidase gene, we have investigated changes in the development and function of the meso-striatal dopaminergic system in MPS-IIIA and their sensitivity to dopaminergic drugs. Using different in vitro models of MPS-IIIA (primary mesencephalic neurons, induced dopaminergic neurons from embryonic fibroblasts and neuroblastoma cell line knocked-out for the gene SGSH by CRISPR/Cas9), we tested the role of HS in regulating dopaminergic dysfunction in MPS-IIIA.

## Results

**Autistic behaviours precede dementia-like behaviours in MPS-IIIA mice.** We used a classic set of behavioural tests to identify autistic-like behavioural phenotypes in mice[20,22], namely locomotion and stereotyped behaviours in the open field and social interaction tests, including the social tube test and a modified version of the three-chamber social preference test (Fig. 1). We used only male mice, as differences in disease progression have been previously reported between female and male mice[23,24]. To dissociate early behavioural from late-occurring dementia-like behaviours, we focused on the analysis on 2-month-old MPS-IIIA and WT littermate mice (Fig. 1, light blue bars) which, based on our previous studies, have not yet developed lysosomal/autophagosome dysfunctions, inflammation and secondary storages[25]. In MPS-IIIA, early autistic-like behaviours are progressively replaced by the loss of all cognitive and motor abilities acquired during childhood, leading to dementia-like clinical manifestations; therefore, we also tested 8-month-old mice (Fig. 1, dark blue bars) to define the shift to dementia-like behaviours.

Two-month-old MPS-IIIA mice were hyperactive in a novel open field test since they showed increased total distance travelled and maximum speed (Fig. 1a, a″) as well as many forms of stereotyped behaviours including self-grooming and rearing (Fig. 1b, b′). Some of these autistic behaviours have been previously reported[26]. Social interaction was significantly impaired in MPS-IIIA mice as evidenced by consistent social retraction in the tube test when they were confronted with an age-matched never-encountered WT (Fig. 1c). In the social novelty preference task, unlike WT littermates, MPS-IIIA mice showed a lack of preference for the social stimulus over the new object (Fig. 1c′) and this was not due to a basal difference in the exploration (Supplementary Table 1) or impairment in odour discrimination (Supplementary Fig. 1a). At this early stage, as expected, MPS-IIIA mice showed no impairments to neuromuscular capabilities (the hanging wire test) or the memory test (the contextual fear conditioning test) (Fig. 1d, e). In contrast, at 8 months of age MPS-IIIA mice became hypoactive in the open field (Fig. 1a, a″) and showed no evidence of significant stereotyped behaviour or social interaction deficits (Fig. 1b, c′), but they showed impaired performance in the hanging wire (Fig. 1d) test. Consistently with our recent evidence in this model[23,27], in the fear conditioning test, there was no difference in the response to foot shock during the training phase. After 24 h, the same animals were exposed to the same context where they had received the foot shock and WT mice increased their freezing time while MPS-IIIA mice did not, suggesting an impairment in contextual fear memory (Fig. 1e). The shift from autistic- to dementia-like behaviours was progressive, as in 6-month-old mice a mixed set of behavioural impairments was observed (Supplementary Fig. 2a–e).

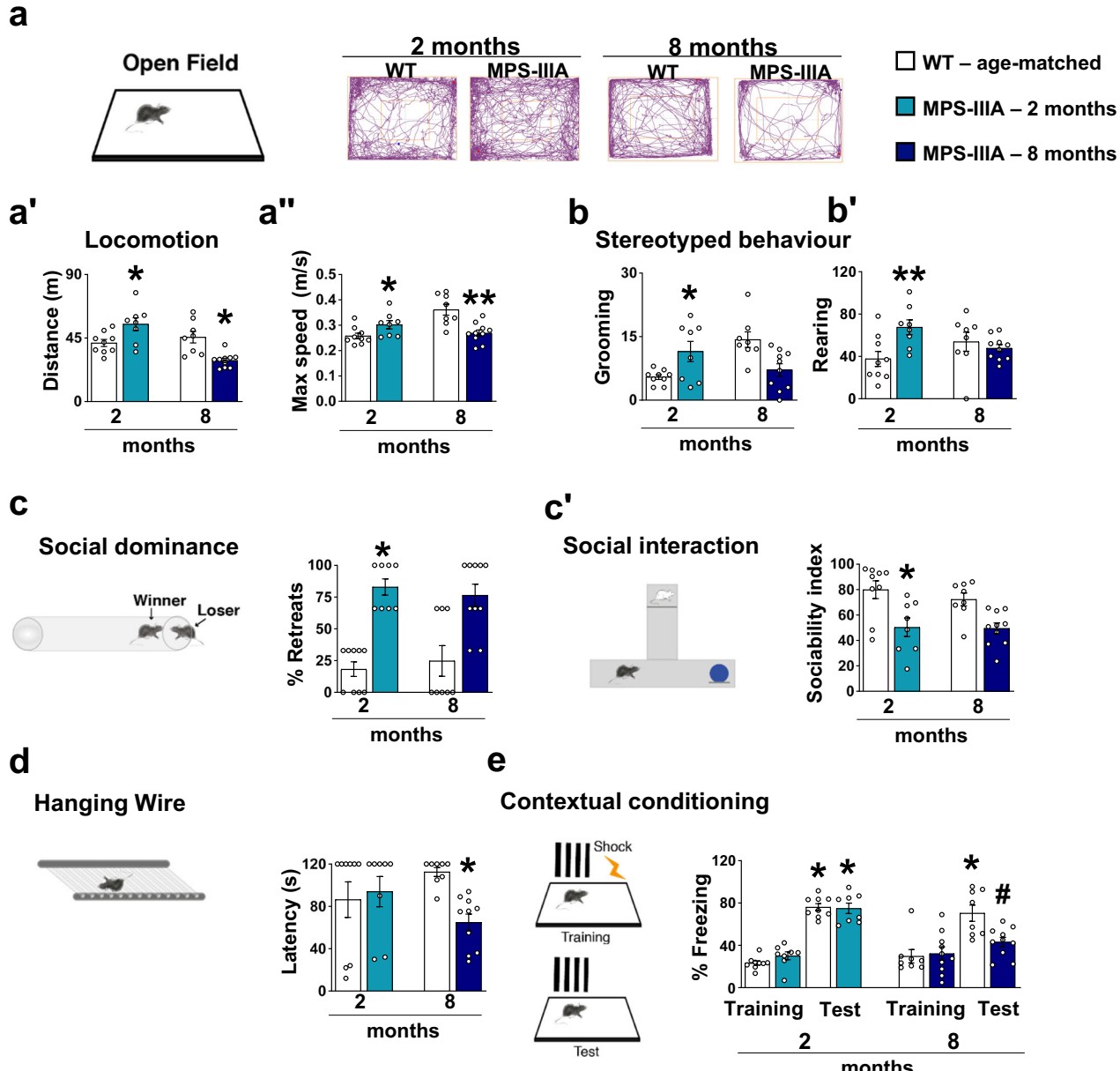

**Fig. 1 Endophenotypes of autistic-like and dementia-like symptoms in MPS-IIIA mice. a** Representative scheme of the open field arena and walking pattern track plots are shown. **a'**, **a"** At 2 months of age, MPS-IIIA (light blue bars) were hyperactive as they showed increases in both the distance travelled and maximum speed, compared to age-matched WT mice. At 8 months of age, MPS-IIIA mice (dark blue bars) were hypoactive as they showed decreases in both the distance travelled and maximum speed, compared to age-matched WT mice. **b**, **b'** 2-month-old MPS-IIIA showed many stereotyped behaviours including self-grooming and rearing and (**c**) displayed strongly impaired social behaviour as evidenced by a higher percentage of retreats when confronted with a WT in the dominance tube test and (**c'**) decreased sociability index in the social novelty preference task. Contrarily (**b-b'**) no differences were detected in stereotyped behaviour and (**c-c'**) in social behaviour in 8-month-old mice compared to WT littermates. **d**, **e** No significant differences were found in the hanging wire and the fear conditioning test between 2-month-old WT and MPS-IIIA mice. However, 8-month-old MPS-IIIA manifested (**d**) neuromuscular impairment in the hanging wire test and (**e**) memory impairment as shown by reduced percentage of freezing time during the contextual fear conditioning test. Histograms represent mean ± S.E.M. $^*p < 0.05$, $^{**}p < 0.01$ vs WT age-matched, between groups; $^{\#}p < 0.05$, vs MPS-IIIA, within group.

These data obtained in 2-, 6- and 8-month-old MPS-IIIA male mice identify age-dependent onset of autistic behaviours followed by dementia-like behaviours. In line with our previous findings[23,25,27], dementia-like, but not autistic-like, behaviours (Supplementary Fig. 3a, a") were concomitant with the age-dependent onset of autophagosome/lysosomal dysfunction as evidenced by increased LC3-II protein expression and an increase of one of its protein substrates, Sequestosome 1 (p62/SQSTM1) (Supplementary Fig. 3b, b").

**Autistic-like behaviour in MPS-IIIA is due to striatal hyper-dominergia and is rescued by the D1-like receptor antagonist.** Hyperactivity, social interaction and stereotyped behaviours are abnormal behaviours subtly regulated by ventral to dorsal meso-striatal and meso-cortical circuits[28,29]. To understand the neurobiological mechanisms leading to autistic-like behaviours in MPS-IIIA and to design specific therapeutic approaches, we focused our biochemical analysis on DA at the level of the corpus striatum (including both the ventral and the dorsal parts), which

is one of the main terminal regions of mesencephalic DA neurons. DA in the striatum acts predominantly on D1-like and D2-like receptors, which are mainly localized on different populations of GABAergic medium spiny neurons and have opposite effects on the 3′,5′-cyclic adenosine monophosphate (cAMP) intracellular signalling pathway, as shown in the scheme of a dopaminergic synapse in Fig. 2a. They are believed to regulate behaviour

by preferentially activating the direct and indirect basal ganglia pathways, respectively. Tyrosine hydroxylase (TH) is the rate-limiting enzyme of DA (for review see[30]) and its expression was increased in 2-month-old MPS-IIIA striatum as compared to that of WT littermates (Fig. 2b). Striatal glutamate decarboxylase 65 (GAD65) (Supplementary Fig. 3c), a marker of GABAergic neurons, was unchanged, suggesting that the increased TH level

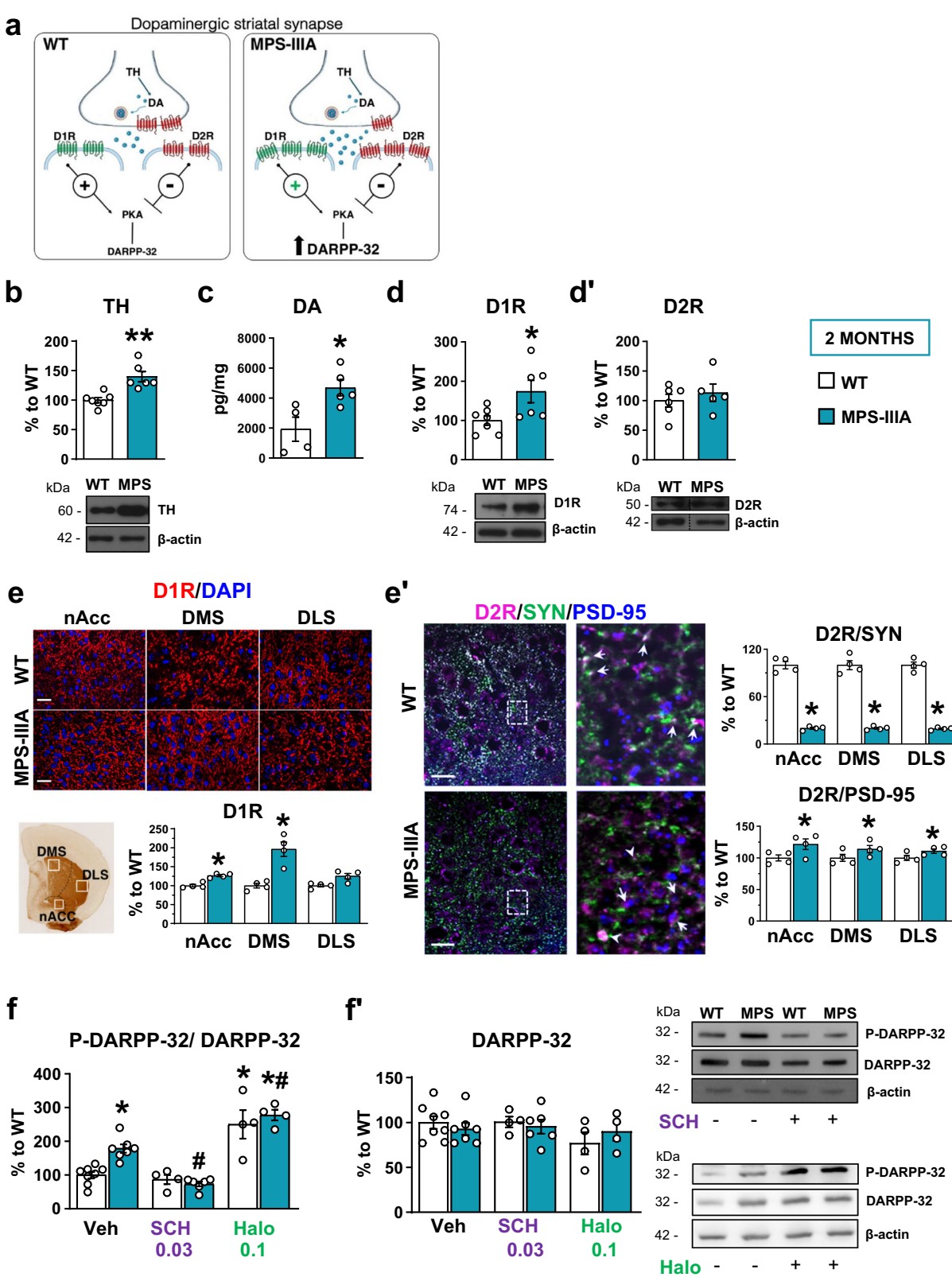

**Fig. 2 Increased expression of D1 dopamine receptors and hyperactivated striatal D1-pathway in MPS-IIIA mice. a** A representative scheme of a striatal dopaminergic synapse, including D1-direct and D2-indirect DA signalling, on the 3′,5′-cyclic adenosine monophosphate (cAMP) intracellular pathway. **b** In the striatum, MPS-IIIA mice showed increased expression of TH, evaluated by WB analysis. **c** MPS-IIIA mice have increased striatal DA, **d** increased expression of D1Rs and **d'** no change in the expression of D2Rs. **e** These data were confirmed by immunofluorescence analysis against D1R. Representative images of D1Rs/DAPI immunofluorescence of the shell of accumbens nucleus (nAcc), the dorsolateral (DLS) and dorsomedial (DMS) striatal areas of WT and MPS-IIIA mice are shown (scale bar: 20 μm). **e'** Triple immunolabeling against D2R, SYN and PSD-95 revealed decreased colocalization of D2R/SYN and increased colocalization of D2R/PSD-95, suggesting decreased and increased expression of D2R at pre- and post-synaptic level, respectively. Representative images of D2R/SYN/PSD-95 immunofluorescence of WT and MPS-IIIA mice are shown. Digital zoom of the boxed area showing coexpression of D2R/SYN (arrows for WT mice, and arrowheads for MPS-IIIA mice) and of D2R/PSD-95 (arrows for MPS-IIIA mice), (scale bar: 50 μm). **f, f'** MPS-IIIA showed hyperactivation of DARPP-32 intracellular signalling, which is rescued by SCH-23390, but not by haloperidol, treatment. Representative WB for each condition is presented. Dashed lines in (**d'**) delineate cut parts from the same western blot. Histograms represent mean ± S.E.M. *$p < 0.05$, **$p < 0.01$ vs WT, between groups; #$p < 0.05$ vs MPS-IIIA veh, within group.

was not a consequence of a generalized increase in protein expression or defective protein degradation. We confirmed, here, that the increased TH expression was functional, since HPLC analysis showed that DA was increased in striatal tissue (Fig. 2c). Biochemical and immunohistochemical analysis of the expression of these two receptor subtypes in 2-month-old MPS-IIIA mice showed an imbalance in the activation of these two pathways. MPS-IIIA mice showed increased expression of striatal (including the nucleus accumbens and the dorsomedial striatum) D1 dopamine receptors (D1Rs) (Fig. 2d, e) while the expression of D2Rs was not affected (Fig. 2d' and Supplementary Fig. 3d). D2Rs, however, are also expressed pre-synaptically where they inhibit the release of DA. Therefore, a lack of change in total D2R expression might not account for an imbalance between the pre- and post-synaptic component. Therefore, we addressed this issue by performing a double immunostaining for D2Rs with pre- and post-synaptic markers, synaptophysin (SYN) and post-synaptic density protein 95 (PSD-95), respectively. We have found that D2R pre-synaptic expression is decreased in MPS-IIIA striatum as compared to WT littermates (Fig. 2e'). By contrast, D2R post-synaptic expression is increased (Fig. 2e'), although this increase is much lower compared to that observed for D1Rs (15% vs 49%, respectively), in the absence of changes in basal expression of SYN and PSD-95 (Supplementary Fig. 3e, f). Thus, reduced D2 pre-synaptic inhibitory action further supports hyperdopaminergia in MPS-IIIA mice, which results in over-stimulation of the D1-direct pathway, as evidenced by increased phosphorylation of its downstream target, the DA and cAMP-regulated phosphoprotein of relative molecular mass 32,000 (p-DARPP-32). The increased phosphorylation was rescued by in vivo administration of the D1-D5R-selective antagonist, SCH-23390, but not by haloperidol, which consistently with previous findings increased p-DARPP-32 in WT animals[31], while leading to blunted effects in MPS-IIIA mice (Fig. 2f, f').

Altogether these findings suggest that DA-D1R direct pathway overactivation (Fig. 2a) might be responsible for some of the behavioural deficits observed in young MPS-IIIA mice. To address this issue, we used a pharmacological approach first by systemically injecting the TH-selective inhibitor, alpha-methyl-p-tyrosine (α-MPT)[32]. Two hours after injection, α-MPT did not significantly affect basal locomotor activity in WT animals, but rescued hyperlocomotion in MPS-IIIA mice in the open field test (Fig. 3a). Similarly, α-MPT rescued stereotyped behaviour (Fig. 3a') and social dominance in the tube test (Fig. 3a''). Although α-MPT efficiently rescued all the behaviours we have tested, including hyperactivity, stereotyped behaviour and the social dominance deficits in MPS-IIIA mice, this drug has no clinical relevance since it leads to progressive time-dependent non-physiological full depletion of DA[32] that reflects its general sedative effects 5 h after injection (Supplementary Fig. 3g). Therefore, we manipulated D1Rs to re-establish a balance in

the two receptor pathways by inhibiting D1R activation. To antagonize the hyperactive D1R pathway, we injected MPS-IIIA mice with the D1-D5R antagonist SCH-23390. We found that acute injection of SCH-23390 did not have major motor effects in WT animals but rescued exploratory activity (the distance travelled and the stereotyped behaviours) in the open field test and social interaction in both the social tube and the social novelty test at both doses (Fig. 3b, b''). We also treated mice with the classical antipsychotic haloperidol. In the open field test, the highest dose of haloperidol (0.1 mg/kg) decreased locomotor activity in MPS-IIIA mice with respect to vehicle treatment (Fig. 3c), but did not improve sociability and stereotypies at either of the two doses used (Fig. 3c', c''). Therefore, if anything, haloperidol has only minor effects on hyperactivity at higher doses. These data provide proof-of-concept evidence that preventing TH activation or blocking D1R activation rescues hyperactivity and social interaction deficits in MPS-IIIA mice, indicating a possible disease mechanism and disease-specific therapeutic approaches for autistic-like behaviour in MPS-IIIA.

**Hyperdopaminergia in MPS-IIIA is due to increased proliferation of DA cells originating during embryonic development.** The striatum receives DA projections from the mesencephalon, therefore we evaluated whether increased striatal TH expression was due to an increased number of DA neurons. 2-month-old MPS-IIIA mice showed a higher number of TH+ neurons in the substantia nigra/ventral tegmental area (SN/VTA) complex associated with an increase in the density of TH+ dendritic arbours in SN (Fig. 4a, a'). To test if the expansion was specific, we evaluated the number of Parvalbumin (PV+) neurons in the SN/VTA complex, and observed a slightly, but not significant, increase in the SN of MPS-IIIA mice (Supplementary Fig. 4a, b'). This specificity is also confirmed when considering the overall number of NeuN+ cells (but not of DAPI+ nuclei, Supplementary Fig. 4c, d), whose increase in MPS-IIIA was evident only when cells double-stained with TH and NeuN were taken into account (Supplementary Fig. 4c, c').

To follow the progression of the TH phenotype in MPS-IIIA, we also analysed 8-month-old MPS-IIIA mice in which the complete impairment in the lysosomal/autophagosome degradation pathway (Supplementary Fig. 3b, b'') apparently did not affect the number of TH+ cells in the SN/VTA complex (Supplementary Fig. 5a, b), suggesting an accelerated age-dependent TH+ cell loss. Double immunostaining of TH/cleaved caspase 3 (cc3), which is an apoptotic marker, showed that most of the TH+ cells expressed both nuclear and perinuclear cc3+ spots (Supplementary Fig. 5d). Accordingly, TH expression was decreased in the striatum, further confirming a progressive loss-of-function of meso-striatal DA (Supplementary Fig. 5c). At 2 months of age, no cells were positive for cc3+ (Supplementary Fig. 5e), suggesting that behavioural symptoms and the TH phenotype progressively

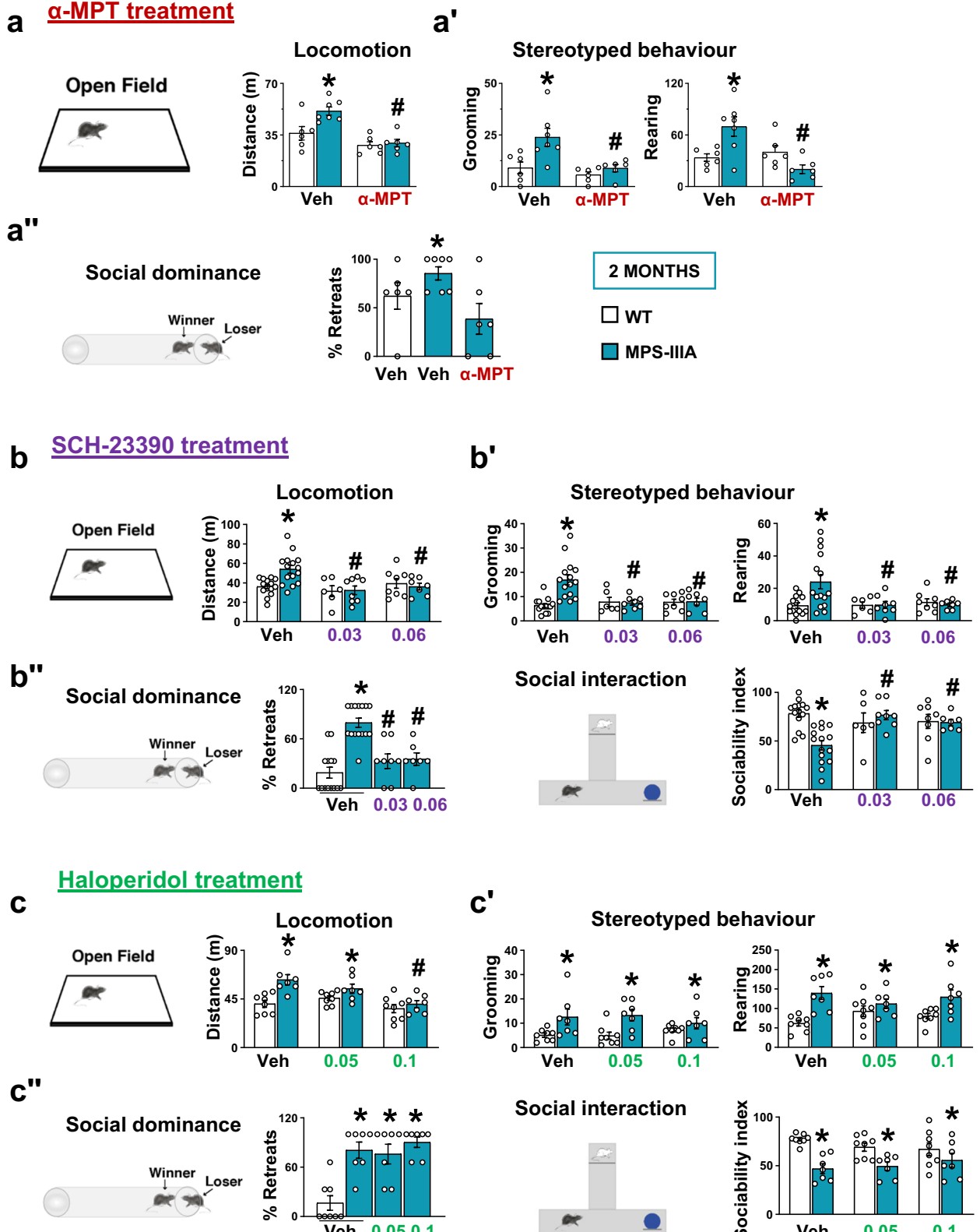

**Fig. 3 The D1 dopamine receptor antagonist, SCH-23390, but not the D2 dopamine receptor antagonist haloperidol, rescues autistic-like symptoms.**
**a** The TH inhibitor α-MPT (250 mg/kg) reduced hyperactivity, **a′** stereotyped behaviour and **a″** the percentage of retreats in MPS-IIIA mice. **b** SCH-23390 (0.03 and 0.06 mg/kg) rescued hyperactivity, **b′** stereotypies and **b″** social impairment. **c** Haloperidol (0.05 and 0.1 mg/kg) only rescued hyperactivity at the highest dose, while had no effect on the: **c′** stereotypies and **c″** social impairment. Histograms represent mean ± S.E.M. *$p < 0.05$ vs WT, between groups; #$p < 0.05$ vs MPS-IIIA veh, within group.

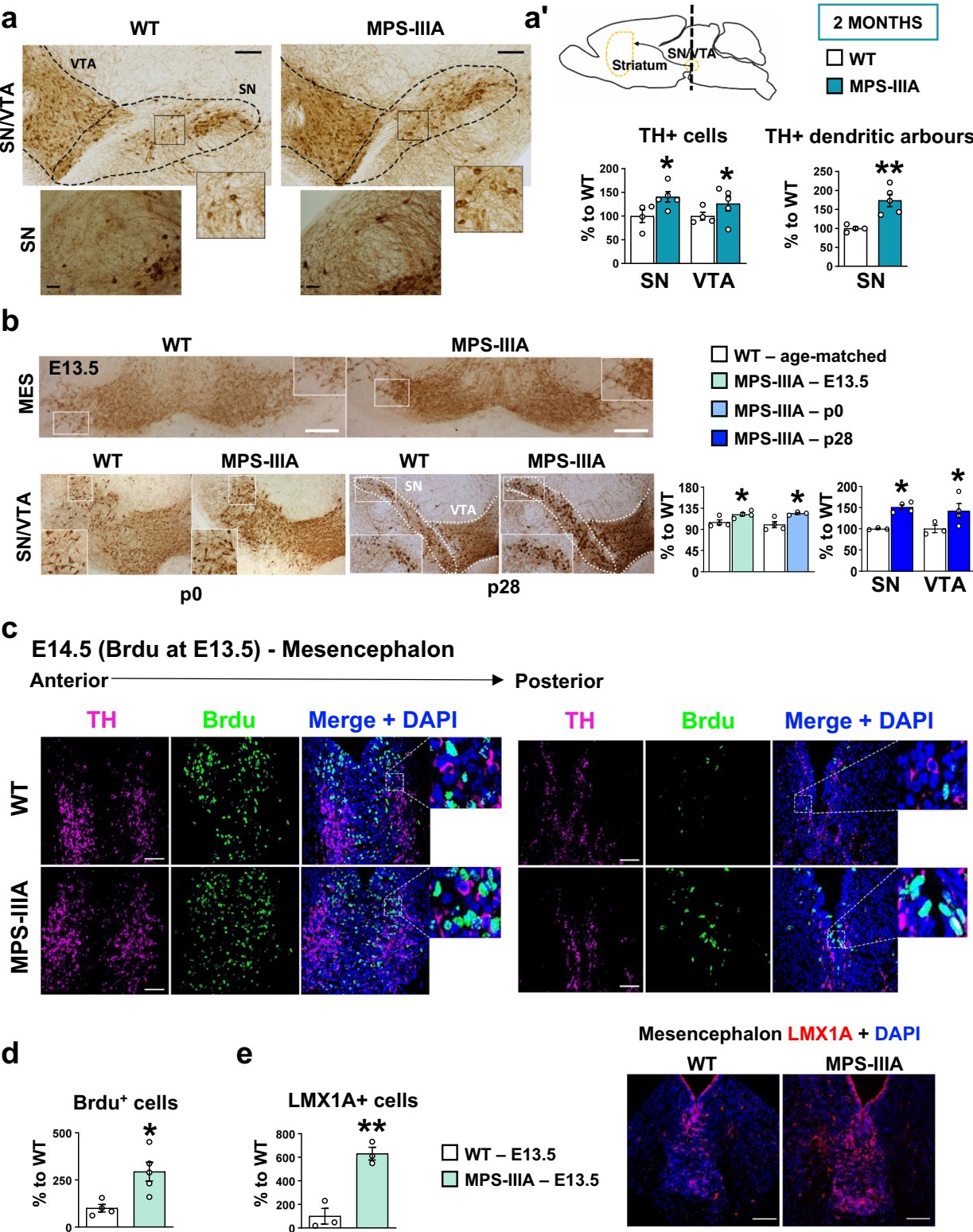

**Fig. 4 Increased proliferation of dopaminergic cells in MPS-IIIA mice begins during embryonic life. a**, **a′** Immunohistochemical analysis of TH+ cells in the substantia nigra/ventral tegmental area (SN/VTA) complex revealed that MPS-IIIA mice had increased number of TH + cells (scale bar: 100 μm) and increased dendritic arbours density in substantia nigra (SN) (scale bar: 25 μm). **b** An increased number of TH+ neurons is observed in embryonic, newborn and juvenile MPS-IIIA mice (scale bar: 100 μm). **c**, **d** MPS-IIIA mice have an increased number of BrdU+ cells in the mesencephalon (scale bar: 20 μm), and **e** increased number of LMX1A+ cells (scale bar: 50 μm). Histograms represent mean ± S.E.M. $^*p < 0.05$, $^{**}p < 0.01$ vs WT, between groups.

worsen in parallel with lysosomal/autophagosomal dysfunction and HS accumulation[23,25,27].

Increased TH expression in young adult MPS-IIIA mice suggests that it may have a developmental origin. To define the developmental trajectory of DA neurons in MPS-IIIA, we counted TH+ cells at earlier time points. In juvenile MPS-IIIA mice (postnatal day 28, p28), we detected an increase of TH+ neurons, which was also evident in newborn MPS-IIIA mice (p0) and at embryonic stage E13.5 (Fig. 4b). To understand if the expanded dopaminergic domain was due to increased proliferation during embryonic development, we performed a 24 h 5′-bromo 2′-deoxyuridine (BrdU) pulse-chase analysis on E13.5 embryos. Interestingly, in the mesencephalon the percentage of BrdU+ cells was significantly higher in MPS-IIIA embryos, compared to WT littermates (Fig. 4c). These data suggest that DA neurons in MPS-IIIA mice have higher proliferation rates. In fact, the number of LMX1A+ cells, a progenitor of DA neurons[33], was increased in MPS-IIIA embryos as compared to WT littermates (Fig. 4e).

These ex vivo findings report an increased proliferation of dopaminergic cells in MPS-IIIA embryos as compared to WT, which is maintained during postnatal life. This initial expansion progresses toward a neurodegenerative histopathological phenotype with the concurrent onset of lysosomal dysfunction.

**Increased proliferation of DA cells in in vitro models of MPS-IIIA.** Increased proliferation of DA cells does not occur in parallel with lysosomal dysfunction; therefore, we hypothesized that it was due to the altered function of HS. To test this hypothesis, we switched to in vitro models of MPS-IIIA. We had previously reported that primary neuronal hippocampal cultures reproduce the time-dependent onset of autophagy/lysosomal dysfunction associated with neuronal loss and synaptic dysfunctions, starting from day in vitro 10 (DIV10)[25]. Here we tested whether they also reproduce the specific increase in DA cell proliferation in the early stages. We first evaluated the number of TH+ cells in WT and MPS-IIIA mesencephalic primary neuronal cultures (Fig. 5). We found an early significant increase in the TH density in MPS-IIIA cells compared to WT (DIV4), followed by no significant differences at DIV7. Conversely, at DIV14, while TH density increased in WT culture, there was a significant decrease in TH+ cells in MPS-IIIA (Fig. 5a, b), coincident with the progressive onset of lysosomal/autophagosomal dysfunction, as increased p62+ and LAMP-1+ spots were detected in MPS-IIIA cells (Supplementary Fig. 6a, c′), associated with an increased density of TUNEL+ cells (Supplementary Fig. 6b, c″). Indeed, while at DIV4 and DIV7 there was no difference in the number of Microtubule-associated protein 2 (MAP2+) neurons, at DIV14 we detected a decrease in MPS-IIIA cells, further suggesting that generalized cell death occurred at this time point (Fig. 5c). We confirmed these findings in two other cellular models. We obtained induced dopaminergic neurons (iDA) from mouse embryonic fibroblasts (MEFs) (Supplementary Fig. 7) by the overexpression of three transcriptional factors: Nurr1, Lmx1a and Mash1[34,35]. iDA cells from MPS-IIIA mice recapitulate the cellular phenotype observed in primary mesencephalic neurons (Supplementary Figs. 7 and 8). By contrast, when WT and MPS-IIIA were infected with a cocktail of transcriptional factors (Ngn2, Myth1, Brn2 and Mash1)[36] to induce the differentiation into pyramidal neurons (iPy), there were no differences in the number of MAP2+ cells at DIV4 between WT and MPS-IIIA cells (Supplementary Fig. 9a, b). However, at DIV14 there was a decrease in the number of iPy neurons in MPS-IIIA (Supplementary Fig. 9a, b), further confirming an increased proliferation

of dopaminergic cells in the early time-point and generalized cell death at later time-points due to autophagy/lysosomal dysfunction. Finally, we used a human cellular model of MPS-IIIA, the neuroblastoma cell line (SH-SY5Y) knocked-out for the gene SGSH by CRISPR/Cas9 technology (called CRISPR/Cas9 MPS-IIIA from now on). The SH-SY5Y cell line has been used to model Parkinson's disease since both differentiated and undifferentiated SH-SY5Y cell lines express TH[37]. We confirmed that our undifferentiated SH-SY5Y cell line expresses TH by immunofluorescence and western blot analysis (Supplementary Fig. 10a, b) and that it releases DA (Fig. 6b). CRISPR/Cas9 MPS-IIIA showed larger nuclei and increased proliferation of TH+ cells leading to increased DA release (Fig. 6a, b). Indeed, although at DIV1 there were no differences between CRISPR/Cas9 MPS-IIIA and WT in terms of TH+ cell density and DA release, at DIV4 a significant increase of both was detected in the CRISPR/Cas9 MPS-IIIA as compared to the WT cell line (Fig. 6a, b). We did not quantify MAP2+ cells as all cells expressing MAP2 also express TH, therefore this would be a redundant measure. We also confirmed this finding with a cell viability assay (MTS) (Fig. 6c). Interestingly, the CRISPR/Cas9 MPS-IIIA cell line also showed an increased percentage of BrdU+ cells, further confirming the increased proliferation (Fig. 6d).

**SH-SY5Y CRISPR/Cas9 MPS-IIIA cell line model shows increased proliferation of DA cells, which is reduced by the application of wild-type functional heparan sulfate.** The signalling pathway of HS modulates both the proliferation and differentiation of stem cells[13,15,16]. Therefore, to understand the role of HS in determining increased DA cell proliferation, we first transfected the CRISPR/Cas9 MPS-IIIA cell line with a plasmid expressing the hSGSH gene under the control of a CMV promoter. The construct contained a myc-FLAG tag at the C-terminus[38]. We found a full rescue of the proliferative phenotype (Fig. 7a, a‴).

To understand if the increased proliferation in CRISPR/Cas9 MPS-IIIA cell line is dependent on HS-signalling impairment, we grew them in a conditional medium with native HS. We found that HS treatment (25 and 40 μg/mL) reduced cell proliferation in a dose-dependent manner in MPS-IIIA cells (Fig. 7b, b‴), without affecting cellular death (Supplementary Fig. 10c, c′), suggesting that the HS-signalling pathway plays a significant role in TH+ cell proliferation. Higher doses affected cell death in the WT cell line (data not shown). We further confirmed these data by treating the CRISPR/Cas9 MPS-IIIA cell line with HS extracted from WT mice (here referred as HS-WT) and obtained similar results (Supplementary Fig. 10d, d′). None of these treatments affected proliferation in WT cells. These data also suggest that in MPS-IIIA brain, as well as in other MPS pathologies, there is altered HS function. In line with previous studies reporting defective HS-FGF2 function in cells of MPS patients[17,18], which is rescued by the replacement of HS from control patients[17], we also found that in assay using Baf32 cells[39] HS-FGF2 signalling was impaired. BaF32 is a lymphoblastoid cell line that lacks cell surface HS and overexpresses the FGF receptor type 1 (FGFR1). These cells proliferate in response to FGF2 only if functional HS is added to cell culture medium. Hence, we compared the capacity of HS extracted from WT and MPS-IIIA mouse brains to induce FGF2-FGFR1-dependent mitogenicity. While HS extracted from WT brain was able to induce FGF2-dependent mitogenicity, HS extracted from MPS-IIIA brain did not (Supplementary Fig. 11a). These in vitro findings show that loss of SGSH gene function leads to altered HS-function, which contributes to increased DA cell proliferation in MPS-IIIA.

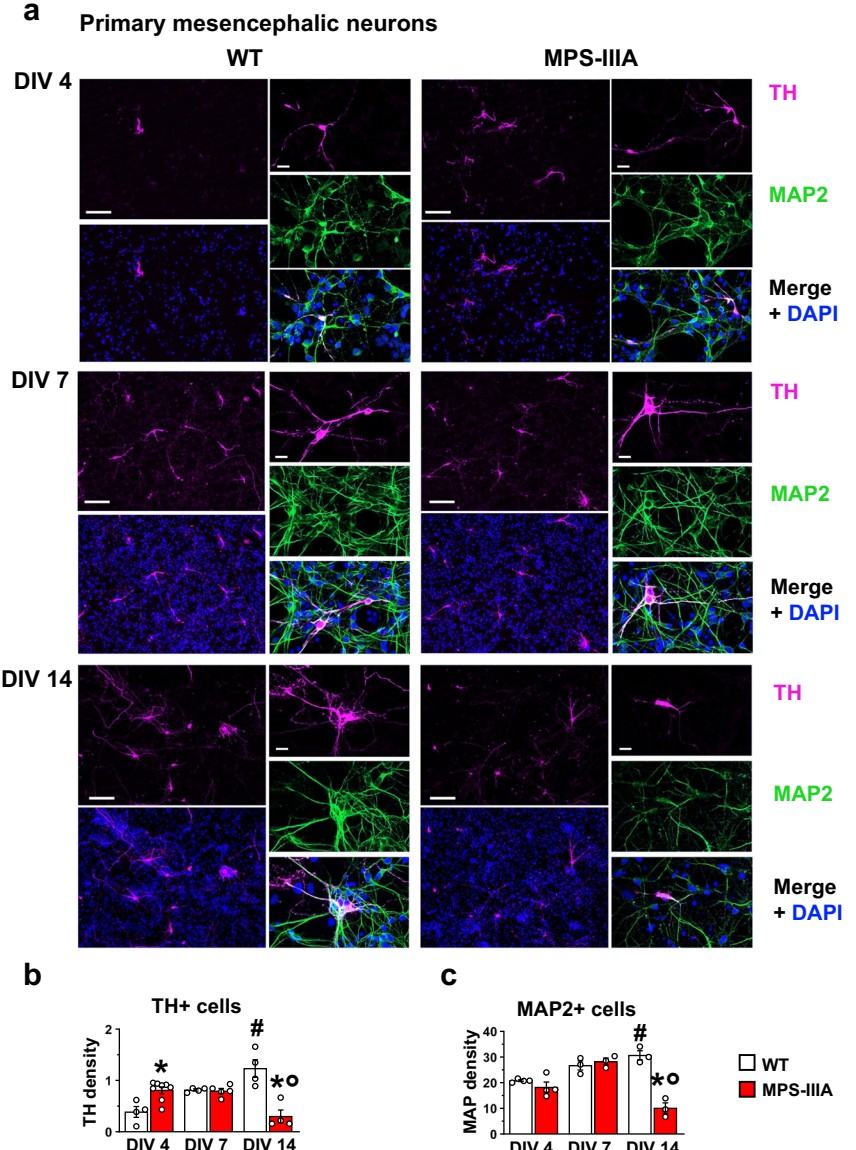

**Fig. 5 Time-dependent increased proliferation of dopaminergic cells in MPS-IIIA mesencephalic neurons. a, b** Primary cultures of mesencephalic neurons reproduce the progression of the TH phenotype in vitro, showing an increase in dopaminergic cell density at DIV4. However, at DIV14, while in WT cells there was a time-dependent increase, MPS-IIIA cells showed a decrease of TH+ cells density (scale bar: 75 µm). **c** MAP2 + neurons were not changed at DIV4 and DIV7, while were reduced at DIV14 in MPS-IIIA cells. Histograms represent mean ± S.E.M. *$p < 0.05$ vs WT, between groups; #$p < 0.05$ vs WT, within groups; °$p < 0.05$ vs MPS-IIIA, within groups.

**Hyperdopaminergia and autistic behaviours also manifest in MPS-II mouse model.** In the MPS-IIIA model we showed that altered HS function leads to hyperdopaminergia and early behavioural symptoms in adult mice. Autistic behaviours occur also in other lysosomal storage disorders with defective degradation of HS, such as in children affected by MPS-II[5,40]. Therefore, we tested an MPS-II mouse model (1-month-old) in which, due to the accumulation of both DS and HS, show faster clinical progression and present lysosomal dysfunction from as early as 3 months[41]. MPS-II mice present no changes in locomotor activity (Fig. 8a, a′) but an increase in stereotyped behaviours in the open field test (Fig. 8b, b′). Interestingly, although they did not systematically retract from the social tube test, they tended to avoid conspecifics in the social interaction test (Fig. 8c, c′). The lack of impairment in the hanging wire test and in the contextual fear conditioning memory test (Fig. 8d, e) confirm that this was an early stage preceding the onset of neuromuscular and memory

impairment, as also suggested by a lack of significant autophagy impairment (Supplementary Fig. 12a, a″). Interestingly, they also showed an increased number of TH+ cells in the SN/VTA complex and increased TH expression in the striatum (Fig. 8f, g′). These data report autistic-like behaviours in juvenile MPS-II mice associated with increased TH expression. Consistent with human studies, their behavioural phenotype is less pervasive than that observed in the MPS-IIIA mouse model.

## Discussion

Targeting autistic-like symptoms represents an urgent unmet clinical need in MPS-IIIA patients, which show social and affective abnormalities, hyperactivity, repetitive behaviour and restricted rituals and routines (such as repetitive motor stereotypies and sensory interests)[3,42]. Some autistic-like behaviours have already been described in MPS-IIIA mice[26], however, a full

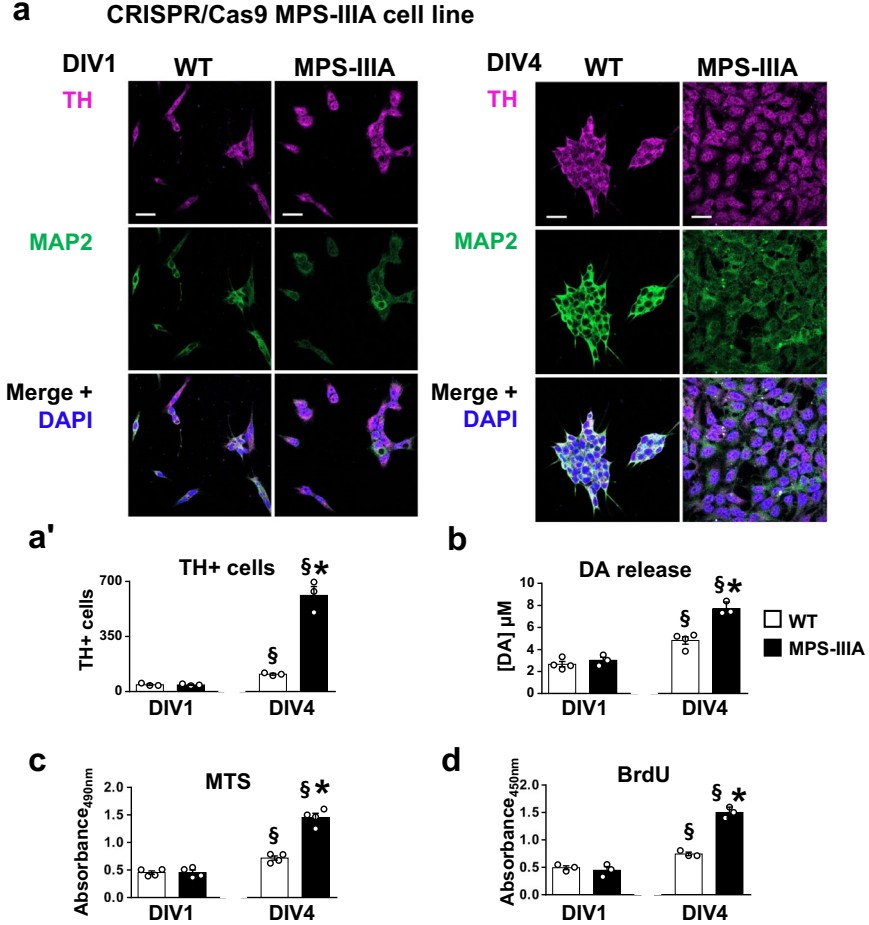

**Fig. 6 Time-dependent proliferation of TH+ cells in CRISPR/Cas9 MPS-IIIA cell line. a** SH-SY5Y MPS-IIIA showed increased cell density at DIV4, compared to WT controls, but not at DIV1. Representative staining for each condition are reported (scale bar: 10 μm). **b** Similarly, while at DIV1 there was no difference in the DA release, at DIV4 MPS-IIIA cells increased DA release. **c** MTS, and **d** BrdU assays confirmed that SH-SY5Y MPS-IIIA cells show an increased proliferation rate, compared to WT cells at DIV4 but not at DIV1. Histograms represent mean ± S.E.M. $^{\S}p < 0.05$ vs DIV1, within groups; $^{*}p < 0.05$ vs WT, between groups.

characterization has not been performed and the disease mechanisms that underpin this disease are yet to be identified.

We expand on these previous observations by characterizing different aspects (motor, social and sensorimotor components) and the time course of the disease. Although the behavioural tests used in this study aim to dissect species-specific behaviours, which therefore can phenotypically differ from the clinical manifestation in humans, they are generally accepted to reflect autistic- and dementia-like symptoms. Additionally, they allowed us to separate early-occurring and late-occurring behavioural manifestations that are associated with the absence and presence, respectively, of autophagosomal dysfunctions, therefore providing evidence that the MPS-IIIA animal model recapitulates progression of the pathology.

In this study we report a selective increase in the proliferation of dopaminergic progenitor cells in MPS-IIIA embryos and TH+ cells in in vitro cellular models. In the latter, increased proliferation is reduced by replacement with functional HS extracted from the brain of WT animals. These findings provide a novel interpretative framework for the existing literature. Indeed, we show that autistic-like behaviours in MPS-IIIA are associated with a gain-of-function of meso-striatal DA, which begins in the embryonic stages and is still evident in young adult animals. Increased DA innervation and reduced pre-synaptic expression of D2Rs, suggesting a further loss of inhibitory control on the release

of DA, results in over-stimulation of the upregulated D1-pathway, in untreated symptomatic MPS-IIIA young mice. These changes in DA signalling in MPS-IIIA might explain why D2-like receptor antagonists, such as haloperidol and risperidone, have little therapeutic effect on this pathology[7–9]. These antipsychotics act as D2-like receptor antagonists, therefore, by further down-regulating D2 signalling they might further stimulate DA release and shift its action toward the D1-pathway (when D2Rs are occupied by the drug), leading to minimal beneficial effects on autistic-like behaviours. Indeed, we show that reducing DA signalling with a TH inhibitor or with the D1R antagonist (SCH-23390) rescued autistic-like behaviours in MPS-IIIA mice, while the D2Rs antagonist produced only a dose-dependent rescue on hyperactivity. Although we focused on DA, we do not exclude that other synaptic re-arrangements might contribute to these behavioural symptoms. Previous evidence reported increased PSD-95 punctae in cortical layers I, II/III and V in 28-day-old MPS-IIIA mice compared to heterozygous ($Sgsh^{+/h}$), which was not observed in 14-day-old mice. These neuronal changes were associated with the accumulation of altered HS and characteristic non-reducing end (NRE) glycanic structures in the absence of a secondary pathology (lysosomal dysfunction, astrogliosys, etc.)[43]. However, they were not causally linked to autistic-like behaviours. Similarly, although we have focused on the striatum (including both the dorsal and the ventral parts) we do not

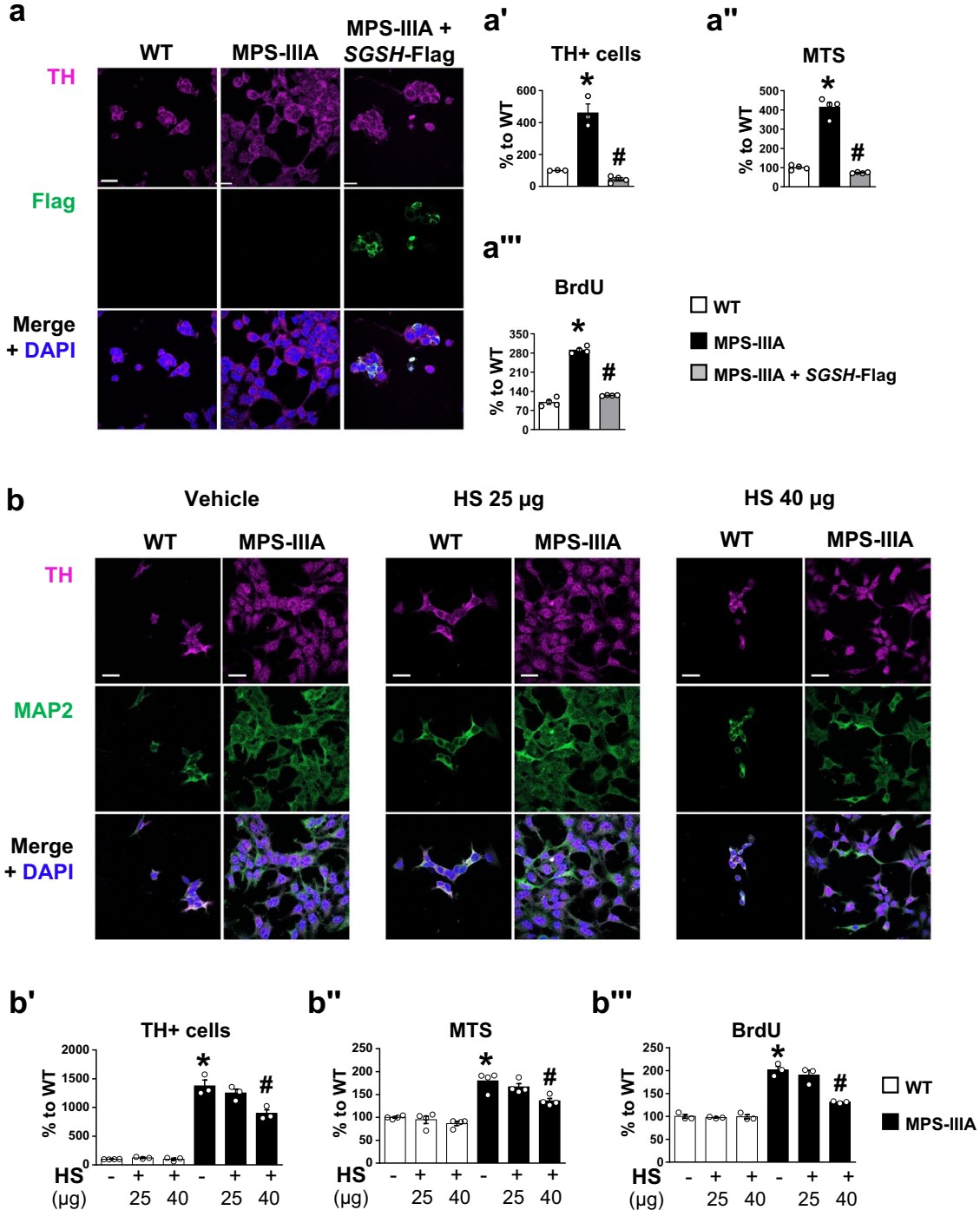

**Fig. 7 Replacement of functional heparan sulfate partially rescued the increased proliferation in the SH-SY5Y MPS-IIIA cell line. a**, **a′** CRISPR/Cas9 MPS-IIIA transfected with a plasmid expressing hSGSH gene with a myc-FLAG tag showed a decreased percentage of TH+ cells, compared to non-transfected cells. **a″**, **a‴** Similarly, MTS and BrdU assay showed a significant decrease in MPS-IIIA transfected cells. **b**, **b′** HS treatment (25 or 40 μg/mL) decreased the number of TH+ cells, and **b″**, **b‴** the proliferation rate, evidenced with MTS and BrdU assays. Representative staining for each condition is reported (scale bar: 10 μm). Data are expressed as mean ± S.E.M. *p < 0.05, between groups; #p < 0.05 vs MPS-IIIA vehicle, within group.

exclude that the effects observed are due to similar mechanisms in other DA terminal regions such as the prefrontal cortex, the amygdala and the hippocampus. Striatal DA is, in any case, relevant for all the behavioural phenotypes we have observed. A recent elegant study demonstrated that optogenetic stimulation of DA release in the striatum of normal mice leads to sociability deficits and repetitive behaviours relevant for the ASD pathology that were rescued by a D1Rs antagonist[44]. Interestingly, in the

same study it is reported that pharmacological activation of D1Rs in normal mice or the genetic inhibition of D2Rs also produced typical autistic-like behaviours that were rescued by D1Rs inhibition. Thus, an imbalance in the direct/indirect striatal pathway could be one of the disease mechanisms responsible for some autistic-like behaviours.

Although these findings have been obtained in an animal model and use standard tests for autism models, they need to find

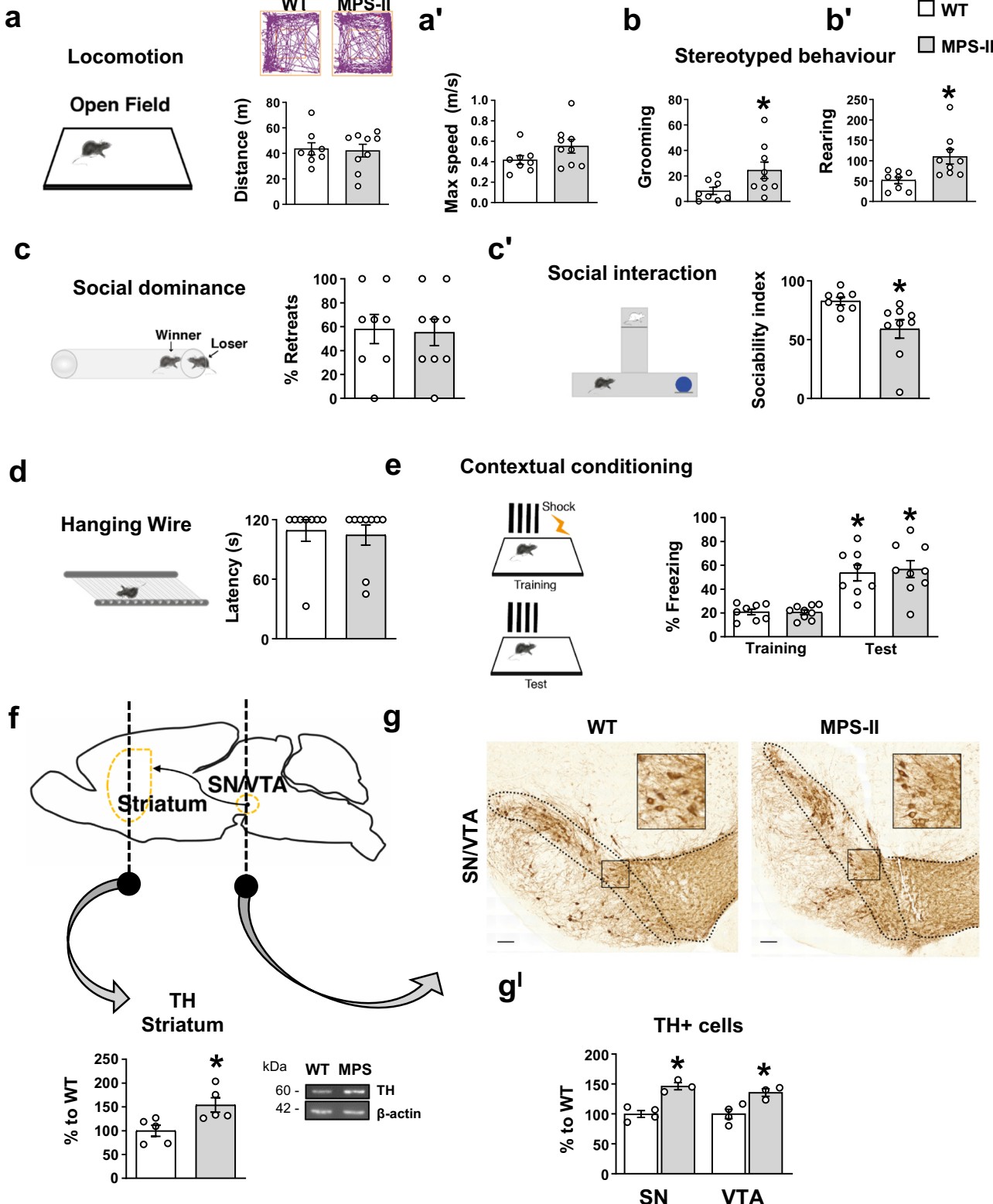

**Fig. 8 Autistic-like behaviours and increased DA cell number are also present in an MPS-II mouse model. a, a'** No locomotion differences were present in the 1-month-old MPS-II mice, compared to WT littermates. Walking pattern track plots are shown. **b, b'** MPS-II mice manifested stereotypies. **c-c'** They did not show an increased percentage of retreats in the social tube test, but presented increased sociability index. **d, e** No significant differences were found in the hanging wire and in the fear conditioning test between WT and MPS-II mice. **f** MPS-II mice, compared to WT littermates, had increased expression of TH in the striatum and **g, g'** showed increased number of TH+ cells in the SN/VTA complex. A representative WB for each condition is presented (scale bar: 100 μm). Histograms represent mean ± S.E.M. *$p < 0.05$ vs WT, between groups.

support in clinical trials. For instance, an analogue of SCH-23390, ecopipam, is being tested in clinical trials for tics in Tourette syndrome[45] and to correct moderate to severe self-injurious behaviours in Lesch-Nyhan patients[46]. Clozapine, an atypical neuroleptic that, unlike D2R antagonists, has proven effective in treatment of self-injurious behaviour is also a D1R antagonist (while acting also on other neurotransmitter systems).

Our findings report a proof-of-concept pre-clinical evidence on the use of D1R antagonists to treat autistic-like behaviours in MPS-IIIA. Although a role of autophagy in autism seems to be supported by recent evidence[47], in our model at this early stage we have no evidence of impaired lysosomal/autophagosomal degradative capacity, consistent with previous findings in the literature[23,25,38]. Furthermore, the autistic-like behaviours and dementia-like stages can be phenotypically dissociated in terms of behavioural patterns (hyperactivity vs hypoactivity; normal learning vs cognitive impairment, etc.) and dopaminergic status (increased TH+ cells vs decreased TH+ cells; increased TH striatal expression vs decreased TH expression, etc.) suggesting that parallel disease mechanisms might concurrently lead to autistic-like and dementia-like symptoms in MPS-IIIA.

These findings raised the question about the molecular mechanisms leading to increased dopaminergic progenitors' proliferation in MPS-IIIA, which we have reproduced here in three different cellular models of the pathology.

HSPG are major constituents of the extracellular matrix and previous studies showed that downregulation of glypican 4, one of the HSPGs, increases differentiation into dopaminergic neurons expressing TH and nuclear receptor related-1 protein (Nurr1) of embryonic stem cells[13], thus providing support for a specific role of HS in DA neuronal development and differentiation. Our findings clearly show that replacement with a functional *SGSH* gene or functional HS reduced the increased TH+ proliferative phenotype in SH-SY5Y cells, thus providing a mechanistic link between HS signalling and DA cell proliferation in MPS-IIIA. These data suggest that HS in MPS-IIIA has a modified chemical structure that, in turn, alters its function as a co-receptor, in line with previous reports showing structural changes in HS from MPS-IIIA brains. In particular, disaccharide analysis showed an overall increase in charge density, with a clear increase in the degree of 2-O-sulfation in MPS-IIIA, notably observable in very young mice[43,48]. The partial rescue in MPS-IIIA SH-SY5Y cells by WT HS suggested that it might compete with the endogenous hypersulfated HS for the binding of growth factors and partially normalize proliferation. Other structural changes, previously reported in MPS-IIIA[43] and MPS-I mouse models[49], could modify the outcome of the binding event between HS-growth factors and receptors[50–52], including the HS-FGF2 signalling, in line with similar findings reported in cells from MPS-IIIB and MPS-I patients[17,18]. Indeed, FGF2 is important for the development and the maintenance of the DA system and FGF2 knockout mice recapitulates the DA phenotype that we have reported in MPS-IIIA and MPS-II mice[53–57].

Although further studies are necessary to address how HS regulates DA cell proliferation in normal and pathological conditions, this set of evidence is in line with accumulating findings demonstrating the crucial role of GAGs, including HS, in the development and maintenance of the nervous system and in autism. Indeed, the Simons Foundation Autism Research Initiative (SFARI) database reports *SGSH* as a syndromic gene for ASD, as evidenced by a study showing that 13 out of 21 children with MPS-IIIA, evaluated with the Autism Diagnostic Observation Schedule (ADOS), meet the ADOS criteria[3]. Accordingly, altered metabolism of HS has been already identified in one of the best characterized mouse models of ASD, BTBR[21], and in the

urine and brain of iatrogenic-ASD patients[58]. Interestingly, an imbalance between D1-D2R pathways has also been reported in BTBR mice[59]. All these findings are also relevant for MPS-II and even if our data are restricted to these two LSDs, they might be also relevant for other LSDs showing altered metabolism of HS, such as MPS-I, where autistic-like behaviour, including social impairment, attention problems and aggressive behaviours have been recently described[5,6,40]. Our results, obtained in valid mouse models of MPS-IIIA and MPS-II, suggest that a general mechanism involving DA signalling may be common to several neurodevelopmental disorders leading to autistic-like behaviours and provide evidence that ASD symptoms originating during development may still be efficiently managed using appropriate pharmacological intervention in young adults.

In conclusion, in this study we show that inborn errors of HS metabolism lead to altered function of HS resulting in developmental abnormalities of mesencephalic DA in MPS-IIIA and MPS-II. This neurodevelopmental defect manifests as autistic-like behaviours and an altered D1 intracellular signalling increase in the striatum during adulthood. Pharmacological inhibition of D1Rs, but not D2Rs, rescues autistic-like behaviours in young adult mice. Thus, any therapeutic, genetic and/or pharmacological approach aimed at correcting MPS-IIIA must take into account the effects on concurrent pathological processes including the changes in HS-mediated signalling function and of lysosomal/autophagosomal-mediated degradative capacities underlying the different stages of the pathology.

## Methods

**Animals**. MPS-IIIA (*Sgsh*$^{-/-}$)[23,25,27,38], MPS-II (*Ids*$^{y/-}$)[41] and respective control littermate WT congenic C57BL/6 mice were used. Animal studies were conducted in accordance with the guidelines and policies of the European Communities Council and were approved by the Italian Ministry of Health. They were group-housed in Plexiglas cages ($18 \times 35 \times 12$ cm) with free access to food and water and kept at a temperature range between 20 and 23 °C and $55 \pm 5\%$ relative humidity. All procedures were performed during daylight hours (between 9.00 a.m. and 6.00 p.m.). Most of the data relative to MPS-IIIA were replicated on mice from three different animal facilities. All experiments were performed by experimenters blind to the genotype and the treatment.

### Experimental study design

*Experiment 1. Behavioural, biochemical and histological characterization of MPS-IIIA and MPS-II.* We tested MPS-IIIA, MPS-II and their WT littermate mice at different time points (MPS-IIIA: at 2, 6 and 8 months of age and MPS-II: at 1 month of age) in a battery of behavioural tasks (see the experimental scheme in Supplementary materials) to assess autistic-like and dementia-like behaviours (related to Figs. 1 and 8, and Supplementary Fig. 2). At the end of the behavioural tasks, mice were used for biochemical (TH, DA receptors, DA content, etc.) and histological characterization (related to Figs. 2b–e′, 4a–a′ and 8f, g′, and Supplementary Figs. 3a–f, 4, 5 and 12). They were perfused with phosphate-buffered saline (PBS pH 7.4) to completely clear blood from tissue. Brains were rapidly removed from the skull and were divided into two equal parts: one half was fixed in 4% (w/v) paraformaldehyde in PBS and used for histological analysis, and the other half was used for biochemical analysis. An 8-month-old WT mouse was excluded from all the statistical analysis, as it resulted an outlier, according to z-score function.

*Experiment 2. Effects of α-MPT on autistic-like behaviours.* WT and MPS-IIIA mice were used to test the effect of α-methyl-para-tyrosine (α-MPT) on autistic-like behaviours. α-MPT (250 mg/kg i.p.; Sigma Aldrich) was dissolved in NaCl 0.9%. It was injected 2 h before starting behavioural tests. Mice were tested first in the social dominance tube test that was immediately followed by the open field test on the same day to avoid the re-injection of α-MPT on the following days (related to Fig. 3a, a″). We selected the social tube test because it has a lower mean duration with respect to the social novelty task, in this way avoiding a longer delay post injection. Another group of mice was used to test the effect of α-MPT in the open field test 5 h after the injection (later time-point) (related to Supplementary Fig. 3g). The dose of the drug and the testing time were chosen based on literature defining the time- and dose-dependent biochemical and behavioural effects of α-MPT[32,60,61]. After the behavioural tests, the mice were sacrificed by cervical dislocation. They were not used for any biochemical or histological experiment, due to the effects of α-MPT on DA depletion.

*Experiment 3. Effects of SCH-23390 and haloperidol on autistic-like behaviours.* Additional groups of WT and MPS-IIIA animals were used to study the effects of SCH-23390 and haloperidol on autistic-like behaviours (related to Fig. 3b, b″, c, c″). The first cohort of mice was treated with SCH-23390 0.03 mg/kg (i.p.). A single animal received either vehicle or SCH-23390 and it was tested in the social dominance tube test and immediately after in the social novelty test. The day after each animal was re-injected with the same treatment and tested in the open field test to measure both locomotor and stereotyped behaviour. Ten to fifteen days apart, mice were re-injected according to a Latin square design[62]. A second cohort of mice was injected with a higher dose of SCH-23390 (0.06 mg/kg) or vehicle and tested with the same experimental design. A third cohort of mice was used to test the effects of different doses of haloperidol (vehicle, haloperidol 0.05 and haloperidol 0.1 mg/kg), following the same experimental design repeated three times (see the experimental scheme in Supplementary materials). SCH-23390 (Sigma Aldrich) was dissolved in NaCl 0.9% and injected 15 min before the behavioural task. Haloperidol (Sigma Aldrich) was dissolved in acetic acid. To obtain the final concentration, it was diluted with bi-distilled water and NaOH to pH 7.4. Vehicle solution was prepared with the same procedure adding the same quantity of acetic acid and NaOH. Animals were injected 30 min before the behavioural task. After the last behavioural test, mice were sacrificed and not used for any biochemical or histological analysis.

*Experiment 4. Effects of SCH-23390 and haloperidol on DARPP32 pathway.* To perform the biochemical analysis on DARPP32 pathway (related to Fig. 2f, f′), a group of mice was injected with the drug or vehicle, submitted to the open field test and then sacrificed by cervical dislocation. Striatal tissues were used for western blot. Two mice injected with SCH-23390 were excluded from the statistical analysis, as they resulted outliers, according to z-score function.

*Experiment 5. Characterization of embryonic and postnatal development of the dopaminergic system.* p28/p0 mice were directly sacrificed and used for TH counting. For embryos, heterozygous female mice were sacrificed. Embryos were collected and then used for immunohistochemistry (related to Fig. 4b–e).

### Behavioural procedures

*Social dominance tube test.* The test was performed as previously described[20]. Subjects of different genotypes (WT/MPS-IIIA or WT/MPS-II mice) were released simultaneously into the opposite ends of a clear, narrow tube (30 cm in Plexiglas). The animal was considered as a "loser" when it retreated out of the tube. We performed three trials and then calculated the percentage of retreats for each genotype with the following formula: [(number of exits or retreats/total number of trials) × 100]. The percentage was analysed using a $\chi^2$ test.

*Social novelty preference task.* Briefly, the apparatus consisted of three arms (37 × 9 cm) in Plexiglas. In the first session (habituation phase), the mouse was left in one of the three arms; two identical objects were positioned in the other two arms. The mouse was left to explore them for 10 min. After habituation, an unfamiliar CD1 mouse matched for age/sex (defined as "stranger") was placed in one of the arms, while a new object was placed in the second arm. The stranger mouse was enclosed in a cylindrical cage which allowed nose contact between the holes. The amount of time spent to explore the stranger mouse or object was recorded by a trained observer, blind to genotype. Social preference (sociability index) was calculated as: 100 × [CD1 exploration/(CD1 exploration + object exploration)].

*Open field test and stereotyped behaviours.* MPS-IIIA and WT mice were left free to explore a Plexiglas arena (35 × 47 × 60 cm) for 20 min. MPS-II and WT littermates were tested in a different Plexiglas arena (43 × 32 × 40 cm). The distance travelled (m) and maximum speed (m/s) were recorded using a video camera (PANASONIC WV-BP330) connected to a video-tracking system (ANY-MAZE, Stoelting, USA). During the open field, stereotyped behaviour (rearing and grooming)[63] was measured by a trained observer, blind to genotype.

*The wire hanging test.* Neuromuscular strength was assessed recording the latency (s) to fall down from a wire, turned upside-down. The cut-off time was of 120 s.

*Fear conditioning test.* Each mouse was trained in a conditioning chamber. It was cubic (30 cm × 24 cm × 21 cm; Ugo Basile) and had a removable grid floor and waste pan. The grid floor contained 36 stainless-steel rods (3 mm diameter) spaced 8 mm centre-to-centre. When placed in the chamber, the grid floor made contact with a circuit board through which a shock was delivered. The shock intensity was 0.5 mA with a duration of 2 s and it was presented for three times and was associated with a context. At 24 h after training, mice were tested without the foot shock but with the same context. Freezing behaviour was defined as complete lack of movement, except for respiration[23,27].

### Immunohistochemistry / immunofluorescence. 
Adult brains have been cut free-floating 50 μm thick sections on a cryostat (Leica), while 20 μm thick sections were cut for brains from p0 and E13.5 mice and processed as previously described[64]. For immunohistochemistry, first, the sections were incubated with primary antibody

directed to TH (1:1000, AB152, Millipore), then, anti-rabbit IgG peroxidase-labelled antibody (1:200, ab6721, Abcam), lastly, they were exposed to an avidin–biotin complex (SP-2001, Vector Laboratories). Signals were developed using 3,3-diaminobenzidine tetrahydrochloride (SK-4100, Vector Laboratories). Images were taken using the slide scanner Axio Scan (Zeiss).

For immunofluorescence, we used the following primary antibodies: cleaved caspase 3 (1:400, 9661, Cell Signaling), TH (1:400, MAB318, Millipore), NeuN (1:500, ABN90, Millipore) and LMX1A (1:1000, AB10533, Millipore). After washing, the slices were incubated with the appropriate secondary antibodies. Images were taken with LMS 700 (Zeiss) at 20x or 40x magnification. For the BrdU assay, BrdU was injected i.p. (B5002, Sigma Aldrich, 100 mg/kg) in E13.5 pregnant heterozygous female mice. At 24 h after injection[57], embryos were fixed in PFA 4%, cut with a cryostat (Leica) at 8 μm thick slices. Slices were incubated with primary antibody against BrdU (1:200, NB500-169; Novusbio) and TH (1:1000, AB152; Millipore). Appropriate secondary antibodies were then used. Images were taken at 40x magnification using a confocal microscope (Zeiss LSM 700).

### TH counting. 
The number of TH+ cells was manually counted on 20x magnification images using ImageJ software (NIH; Bethesda, MD), by an experimenter blind to the genotype. For each experimental group, regularly spaced 50 μm sections were chosen at multiple rostrocaudal levels in order to sample the SN and VTA. Specifically, we sampled considering the following anteroposterior coordinates relative to the bregma (according to Franklin and Paxinos, 2001): −2.92, −3.16, −3.28; −3.52; −3.64. Each experimental group sections (WT or MPS-IIIA and WT or MPS-II) were matched for each coordinate. Cell numbers from each section were averaged and expressed as percentage of WT, as previously done[64,65]. The identification and the definition of subregions (SN and VTA) were done based on the mouse brain atlas (Franklin and Paxinos, 2001) and on cytoarchitectonic characteristics of the two areas (i.e. cell orientation)[66,67]. We delineated the area representative of SN and VTA in the main figures. We used the same methodology for immunofluorescence analysis on TH+, NeuN+ and PV+ neurons. For p0, regularly spaced 20 μm sections were similarly chosen. We counted within the entire area delimited by DAB staining.

Similarly, for E13.5 mice, we performed cell counting as described above, by counting all slices, regularly spaced 20 μm, between the coronal section 18 and the coronal section 21 slices following the prenatal mouse brain atlas Uta Schambra (2008). For these slices, we counted the cells present in the entire area delimited by the DAB staining. We used the same methodology for the BrdU and LMX1A counting.

### TH dendritic arbours quantification. 
We have evaluated TH dendritic arbours immunoreactivity by measuring the optical density in three squared boxes, positioned at different points of the SN. Quantification was done on three regularly spaced 50 μm bilateral sections (corresponding approximately to bregma −3.08; −3.52; −3.64). Optical density values were corrected by subtracting OD from a non-stained area of the same slice and expressed as percentage of the WT from the same immunostaining batch to correct for differences in DAB immunoreactivity[68].

### D1Rs-D2Rs immunofluorescence. 
Brain slices containing dorsal or ventral striatum of WT and MPS-IIIA mice (2-month-old) were processed for immunofluorescence by overnight incubation with D1R (1:200, sc-33660; Santa Cruz) or D2R (1:100, ab5084p, Merck), synaptophysin (1:500, 101004, Synaptic System) and PSD-95 (1:100, 124011, Synaptic System). Immunofluorescence was revealed by specific Alexa-488 or -546 or -350 secondary donkey anti-IgGs (1:500, Invitrogen Life Technology). D1R immunolabeled sections were washed and finally counterstained with DAPI (4′,6-diamidino-2-phenylindole; Sigma-Aldrich, Milan, Italy). Immunofluorescence was analysed by confocal microscopy (Nikon Eclipse Ti2) and the images acquired with a digital camera DS-Qi2 (Nikon) and processed by Image analysis software NIS-Elements C (Nikon, Florence, Italy). The quantitative analysis of the relative abundance of D1R or D2R and colocalization between D2R with SYN and/or PSD-95 was performed per μm² of coronal sections of Caudate (DMS), Putamen (DLS) and shell of accumbens nucleus (nAcc) brain areas of WT or MPS-IIIA mice by using a Leica Metamorph imaging software (Leica Meta-Morph, Germany) according to the method reported in[69]. In brief, the density of distribution of D1R or D2R/SYN/PSD-95 multiple labelled puncta was counted in a region of interest of 4 × 10⁴ μm² for DMS or DLS or nAcc brain area through 5 μm depth thickness by considering $n = 200$ serial Z-stacks (0.5 μm each) per region ($n = 10$ serial brain sections/mouse).

### Western blotting and HPLC analysis. 
Samples were homogenized in RIPA buffer (50 mM Tris-HCl pH 7.4, 1% TX, 150 mM NaCl, 2 mM EDTA and protease/phosphatase inhibitor cocktail). Then, 20 μg of homogenate were analysed by western blot (WB) using standard procedures. Membranes were incubated overnight at 4 °C with primary antibodies: TH (1:1000, AB152, Millipore), D1R (1:500, sc-14001, Santa Cruz), D2R (1:500, AB5084P, Millipore), DARPP-32 (1:1000, AB10518, Millipore), P-DARPP-32-Thr34 (1:1000, AB9206, Millipore), LAMP-1 (1:500, sc-19992, Santa Cruz), p62 (1:500, H00008878-M01, Tebu-bio), LC3B (1:1000, NB100-2220, Novus-Bio) and GAD65 (1:1000, BK3988S, Cell Signaling). β-actin (1:5000, MAB1501, Millipore) was used as housekeeping. Immunoreactivity

was detected by chemiluminescence and bands quantified by densitometry using ImageJ software. We reported normalized data to the WT group because we calculated the percentage relative to the control for each blot, to correct for any type of differences relative to different gels (for example, recycling of primary antibody or time to ECL exposure). HPLC was performed as previously described[64]. Tissue levels of DA (pg/mg wet weight) were used for statistical analysis.

#### Cell culture experiments

*Primary mesencephalic neurons.* Cells were dissociated from embryonic mesencephalon. Briefly, the tissues were enzymatically dissociated by incubation for 30 min at 37 °C in Earle's balanced salts solution containing Papain (Warthington, 20 U/mL), 1 mM EDTA, 1 mM cysteine and 0.01% pancreatic deoxyribonuclease. After addition of 1 mg/mL of bovine serum albumin (fraction V, Sigma) and 1 mg/mL of ovomucoid (Sigma), the cell suspension was centrifuged 10 min at 800g. Then, cells were plated in 24-multiwell plates (Costar, Milan, Italy) coated with 15 μg/mL of poly-D-Lysine (Sigma). Cells were grown in Neurobasal medium (Invitrogen, Milan, Italy), L-glutamine (0.5 mM, Sigma) and B27 (Invitrogen, Milan, Italy). At DIV4, 7 and 14, the cells were fixed in PFA 4% and stained for TH, MAP2, p62, LAMP-1 and TUNEL. Images were taken using a confocal microscope (Zeiss LSM 800) and counted using ImageJ software.

*SH-SY5Y culture.* The SGSH-ko SH-SY5Y (here called SH-SY5Y MPS-IIIA) cell line was provided by J. Monfregola (TIGEM). They were maintained in culture with Dulbecco's modified Eagle's medium and Nutrient Mixture F-12, supplemented with 10% FBS (EuroClone) and 1% penicillin/streptomycin. HS was purchased from Sigma Aldrich (H7640). It was dissolved (25 and 40 μg/mL) in growth medium before use. Before immunofluorescence, BrdU or MTS experiments, cells were plated at a density of 20,000/cm$^2$ in 24-well coated with 15 μg/mL of poly-D-lysine (Sigma).

*In vitro BrdU and MTS assays.* The assays were performed according to the manufacturer's instructions (G3582 Promega and 6813 Cell signalling). Cells were incubated with a BrdU solution for 24 h.

*SGSH transfection.* Cells were transfected using Lipofectamine 3000 (Invitrogen) according to manufacturer's protocols.

*Immunofluorescence.* Cells were fixed in PFA 4%. After washing, they were permeabilized in Triton-X100 0.3% and the blocked in BSA 3%. After incubation of primary and appropriate secondary antibodies (Alexa Fluor, Invitrogen) cells were stained with DAPI (Sigma Aldrich). Acquisition was taken using a confocal microscope (Zeiss LSM 800). For TUNEL, the staining was performed according to the manufacturer's instructions (Cat. No 12156792910, Roche).

*HS extraction from mouse brain.* Frozen brains were homogenized and suspended in a buffer (50 mM Tris, pH 7.9, 10 mM NaCl, 3 mM MgCl$_2$ and 1% Triton X-100) at 4 °C. Samples were then treated with proteinase K to digest proteins (5 μg/mL; Merck) at 56 °C overnight followed by heat-inactivation at 90 °C for 30 min. DNase (7.5 mU/mL; Qiagen) was added to samples to digest DNA and samples were incubated overnight at 37 °C. Then, peptides were eliminated by precipitation with trifluoroacetic acid (TCA, final 10%) at 4 °C followed by centrifugation (13,000g, 20 min). Supernatants were washed with chloroform (x2) to clear TCA and lipids, followed by dialysis to eliminate residual peptides from PK digestion, oligonucleotides from DNase digestion, and many other small molecules (Slide-A-Lyzer Mini Dialysis Units 3 500 MWCO; Pierce). After freeze drying, samples were dissolved in water. Chondroitinase ABC (Chase ABC; Sigma-Aldrich) was used to selectively obtain HS. HS were quantified using Blyscan Sulfated Glycosaminoglycan Assay (B1000, Biocolor). The assay was performed according to the manufacturer's instructions. The absorbance was monitored at 656 nm using a microplate reader (Promega).

*Dopamine detection.* For SH-SY5Y and SH-SY5Y MPS-IIIA, DA levels were measured using a three terminal organic electrochemical transistor (OECT). These devices are made of a gate and a channel where the active material is an organic conductive polymer known as poly(3,4-ethylenedioxythiophene) polystyrene sulfonate (PEDOT:PSS). The fabrication of the device has been reported earlier[70]. In brief, the gate and channel ($W = 2.5$ mm, $L = 0.9$ mm) areas were obtained through photolithographic patterning using Parylene-C as mask and spin coating PEDOT:PSS on the surface; the following peel-off of the Parylene, left PEDOT:PSS only in the photolithographically defined channel and gate areas. Cell medium was collected after DIV1 and DIV4 and placed onto the OECT: the application of a positive bias at the gate ($Vg = 300$ mV) causes the oxidation of DA and the resulting modulation of the channel current is proportional to the DA concentration in cell medium. Interpolating these data with the calibration curve (see Supplementary methods), the DA concentration for each cell line at different time points was calculated.

**Statistical analysis.** The results were expressed as mean ± S.E.M. Statistical analyses were performed using Statview 5.0 (SAS Institute Inc., North Carolina, USA) and Statistica 10.0 (Statsoft, Tulsa, OK, USA) software.

The statistical significance was assessed using one-way ANOVA, two-way ANOVA or repeated measures ANOVA, followed by the Tukey's post hoc test if appropriate. Before applying the ANOVA tests, data distribution normality was tested with the Kolmogorov-Smirnov test. Outliers were evaluated with $z$-score function. Social dominant tube test was analysed using $\chi^2$ test. The number of animals used for each experiment and statistical significance are reported in Supplementary Tables 2 and 3. All the in vitro experiments were performed at least in triplicate. Significance was set at $p < 0.05$.

**Reporting summary.** Further information on research design is available in the Nature Research Reporting Summary linked to this article.

### Data availability

Data are available from the corresponding author on reasonable request. Simons Foundation Autism Research Initiative (SFARI) database was consulted (https://gene.sfari.org/database/human-gene/). Source data underlying the main and supplementary figures are provided as a Source Data file.

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

## Acknowledgements

The authors would like to thank Prof. Paolo Calabresi, Dr. Cornelius Gross, Prof. Enrico Cherubini, Prof. Nicola Brunetti-Pierri, Prof. Antonella De Matteis, Dr. Michele Studer, Dr. Cathal Wilson and Dr. Graciana Diez-Roux for critical revision and discussion of the manuscript; Dr. Cathal Wilson and Dr. Phoebe Ashley-Norman for English editing; Dr. Minh Bao Huynh for the helping with BaF32 cell line. F.S. and C.L. thank Scott Keene (Stanford University) for the material deposition of the OECTs. We are also grateful to all the other members of De Leonibus' lab for helpful discussions and comments on this manuscript. We thank TIGEM's core facilities (Animal House and Microscopy) for supporting this work. This work was supported by grants from the Sanfilippo Children Foundation, Cure Sanfilippo foundation and National MPS Society to EDL and PRIN (2017T9JNLT_004) to GCB.

## Author contributions

M.D.R. conceived and performed experiments, analysed the data and generated a first draft of the manuscript; M.T. performed behavioural and histological experiments and analysed the data; F.G.A., M.G.F. and Y.G. performed ex vivo and in vitro the experiments; J.M. generated and characterized the SH-SY5Y KO cell line; E.M. helped to perform behavioural experiments; S.P. helped with cell culture experiments; L.T. and L.C. performed and analysed D1 receptors immunofluorescence analysis; C.L. and F.S. performed DA release experiment on SH-SY5Y; G.C.B. supervised the neurodevelopment experiments; A.F. provided expertise and materials with the MPS-IIIA mouse model and A.B. with SH-SY5Y cell lines; A.S. and Y.T. provided devices for DA detection; G.T., D.P.G, G.C.B., A.B. and A.F. contributed to the writing of the manuscript; E.D.L. conceived and supervised the study and the experiments, analysed the data and wrote the manuscript.

## Competing interests

The authors declare no competing interests.
