## [Peer Review File. · Nature Communications]

Reviewer #1 (Remarks to the Author):

In this study, De Risi et al have explored a potential novel cause for the autistic-like behavioral changes seen in the lysosomal storage disease: mucopolysaccharidosis type III (Sanfilippo syndrome). Using a previously well validated mouse model with loss of function of the Sgsh gene, they find that these behaviors are related to hyperdopaminergia, in turn apparently due to developmental increases in dopamine producing cells of the midbrain, and can be ameliorated with drugs that affect the dopamine pathway. These behaviors form a very significant source of morbidity and distress for individuals with MPSIIIA and their carers, so identifying a new affected pathway and potential for treatment would be of significant clinical benefit, and the results of this study will be of interest to the broad community. The authors have used a wide range of approaches to investigate their hypothesis which overall I find broadly convincing. I think the findings of this study have the potential to be exciting, however I have a number of comments on the manuscript.

Major comments

A key argument made in this study is that there is an increase in the D1R pathway, with unaffected D2Rs. The authors use either a D1R antagonist or a D2R agonist – Aripiprazole – to rescue some of the autistic-like behavioral changes seen in the Sgsh^{-/-} mouse, either way ‘rebalancing’ the D1/D2 activity. My issue here is that Aripiprazole is a D2R partial agonist – and is often thought to decrease D2 activity (at least in the mesolimbic pathway) as it is less active than dopamine itself. While it may well be more complex than this, I think for me this section would be much more compelling if the authors could show that, for example, classic D2R antagonists such as haloperidol or risperidone do NOT have the same beneficial effects.

Related to this, the authors used α -MPT to rescue some behaviors, but this drug is known to cause hypoactivity at lower doses than used here (Khakimova et al, Mol Neurobiol 2017). Can the authors include its effect on the social interaction test – this would be the better test.

Another really important part of this manuscript is the quantification of increased numbers of dopaminergic cells (using TH staining), and at several developmental stages. However, the methodology used could be improved and the details given are insufficient. Overall, given the different approaches (including HPLC, Western blots, etc.) I am convinced that the key finding of increased activity in this pathway is probably true, but they could be more compelling. It would be preferable to use fluorescence immunolabelling and double label with at the very least DAPI, and ideally a neuronal marker like NeuN. It would be nice to see a comparison with other cell types in the same region (as opposed to hippocampus NeuN which is unrelated) and to confirm whether overall cell numbers are changed. As all data are presented as normalized values, it is not clear whether these are numbers of cells or densities of cells (both would be good), nor whether they come from regions of interest (ROIs) or the entire region. If the former, how was ROI location chosen? If the latter, how was this delineated? Additional details on quantification are needed – were experiments performed and analyzed blind, were data from sections averaged? The description of the densitometry is unclear and needs more detail (would not be able to reproduce), and it is not clear to me how this accounts for variability in DAB staining per se.

The in vitro cell culture section nicely complements the in vivo work, and overall is convincing. However, I am surprised by the finding of increased TH cell number at 4DIV but not at 7DIV. This appears to not make sense, as if there was higher proliferation early on, there should still be more cells 3 days later. Is there any evidence of cells dying or of lysosomal degradation at this stage (given

that MAP2 seems unchanged, no generalized apoptosis)? Also, they should quantify the total number of cells (eg using DAPI) and give density values, as density effects can be pronounced in culture. Can the authors comment on why there is such a discrepancy with in vivo timing of proliferation and cell death?

Minor comments

In the introduction (line 94) the authors say '[haloperidol 90 and risperidone] ... have considerably higher than expected extrapyramidal side effects in these patients' but ref 7 shows risperidone is effective with no extrapyramidal side effects. While I appreciate this sentence is ambiguous, I think the authors should be more circumspect in their claim here.

Behavior –

No direct test of olfaction was done. This either needs to be added, or if this has previously been done (in their hands preferably) please comment and cite.

Some behaviors in the methods lack a statement as to whether done / scored blind.

Can the authors comment on the repetitive behaviors they see? – these are not a feature of individuals with MPSIIIA, nor are seen in the previous published study of the Sgsh^{-/-} mouse.

Aripiprazole is already licenced for treatment of irritability in autism, but can have many side-effects and sometimes exacerbation of symptoms. While this does not invalidate the findings here (this might be a good example of how to determine who would benefit), the manuscript would benefit from a discussion of this.

This is entirely personal preference, but I think inventing unnecessary new acronyms is best avoided. 'ALBs' is not a standard acronym and not one I have ever encountered, and I would prefer not to use it. Also – I think it is unhelpful, as it is better to refer to the specific behavior.

Social dominance – please give more details. 10cm is very short (eg the reference given uses 30cm).

Why are all the percentages 33 66 100 for WT and different for mutant? Also – please call 'social interaction' social dominance.

Social novelty – at 8 months, seems that the WT aren't interested in stranger mouse, as opposed to any change in the mutants.

Dopamine/TH analysis –

Line 187 – state age (is on figure, but would be good in text too).

Please give more details of D1R analysis in methods – just says used Metamorph.

Why pool data from SN and VTA?

Please outline key regions (eg SN, VTA) on images (and to indicate exactly where quantifications done from)

Line 233-6 – why state that this suggests an accelerated cell loss? Surely going from more cells to normal is expected in this context?

Line 239-40 - 'At 2-months of age, no cells positive for cc3+ (Fig. S2f), suggesting that behavioral symptoms (ALBSs and DLSs) and the TH phenotype progressively worsen in parallel with the lysosomal/autophagosomal dysfunction and HS accumulation' – not clear to me what is being said here. Surely, there is progressive loss of many cell types, so the hyperdopaminergia is gradually lost (indeed, initially returned to normal levels?), while the loss of other neurons leads to dementia? Fig 4c (BrdU) – looks like picked non-representative spot for high power view, at least for anterior mesencephalon.

Fig 7 – no scale bars (and are MPSIII cells as same magnification as WT in DIV4 images? (Nuclei size looks different).

Line 313-5 – I think it is a stretch to say this suggests that there is a loss of HS function in the brain in MPSIIIA (nor other MPS disorders), although this would be very interesting to test directly, if possible. If not, a more nuanced phrasing would be more appropriate.

Fgf2 experiments – this needs a bit more explanation – please briefly justify rationale for this experiment in the results section as well as discussion, please include in methods (currently completely missing) and state what Baf32 cells are.

Please include full Western blots in the supplementary information.

It is very difficult to see scatter plot over dark blue bars, please use different colour/line combination.

Minor corrections

Fig 2A – what is the picture of? Not mentioned in legend.

Line 77 – ‘incapacitating’ instead of ‘invalidating’?

Line 96 – reference 8 does not look at dopaminergic cells – only colocalization of NCAM with HS in layer 5 cortex.

Lin 107 – neurexin

Line 183 – GABA’ergic’

Line 503 – one half, not each half

Line 239 – no cells *were* positive...

Fig 4b – missing scale bars for P0 and P28

Line 269 – ‘autophagosomal’

Fig 5c legend ‘ MAP2+ neurons were *not*

Line 313 – actually Fig S6 not S5

Line 350 – ‘which is rescued’ – need to be clear that have only rescued in vitro effects, not those in embryo.

Reviewer #2 (Remarks to the Author):

The question asked in this work - what is the molecular mechanism of autistic-like symptoms in mucopolysaccharidosis (MPS) type IIIA - is excellent and important to understand the disease. On the basis of a series of well-conducted experiments, the authors demonstrated developmental dopaminergic abnormalities in the MPS IIIA mouse model which were partially confirmed in the MPS II mouse model. Experiments with cellular models of MPS IIIA suggested that there are changes in heparan sulfate (not only accumulation of this glycosaminoglycan) which may contribute to observed biochemical abnormalities. This report can be a significant contribution to the field, indicating previously unknown molecular mechanisms of the disease. Nevertheless, I have several comments which I suggest should be addressed.

Major points:

1. The title suggests that "altered metabolism of heparan sulfate (HS) leads to developmental dopaminergic abnormalities responsible for autistic-like symptoms in lysosomal storage disorders". However, only MPS IIIA and MPS II models were studied. Therefore, I suggest to restrict the conclusions, as well as the title, to these two diseases.

2. The problem of too general conclusions regarding the effects of HS on dopaminergic abnormalities

and resultant autistic-like symptoms (see point 1) arises from the fact that no (or only little) autistic-like symptoms are observed in patients suffering from other diseases in which HS accumulation (and changed metabolism) is evident, like in MPS I. This should be discussed in the light of the proposed mechanism.

3. Undegraded (or rather partially degraded) HS is stored in MPS IIIA cells. However, addition of normal HS or HS derived from WT organisms partially restored selected defects (Fig. 8 b and b'). From the discussion it appears that the authors suggest that HS from WT and MPS organisms differs considerably. Therefore, chemical structures of HS derived from WT and MPS IIIA animals should be determined to indicate if there are significant chemical changes between these molecules which might be responsible for their functional modulations.

4. Although behavioral tests used in this work are generally accepted as reflecting ALBS in mice, it is necessary to indicate that a mouse model was used in which autism might be, nevertheless, quite different than in humans. It is true that there are no better models, thus, the experimental system is fine, but the limitations in interpretation of the results should be discussed.

5. Excess of normal HS did not completely rescue the measured parameters shown in Fig. 8 (panels b and b') but rather improved them towards values estimated for WT controls. This should be indicated in the text, and interpreted not as correction (rescue) but as improvement. As a consequence, HS dysfunction should not be presented as the cause of the observed biochemical and behavioral changes, but rather its contribution to the pathomechanism can be suggested.

Minor points:

6. L.107-108: Neurexin and neuroligin are not genes. There are proteins, encoded by corresponding genes. The authors should clearly distinguish whether they refer to genes or proteins throughout the text.

7. L.110-127: Description of the results of this study is too long in Introduction. At the end of this chapter, rationale of the study should be presented rather than quite long description of obtained results and conclusions.

8. L.222: "indicating a first causal disease mechanism" is a too strong statement. ALBS is a complex syndrome, thus, the interpretation of the results can be presented as either suggestion or possible contribution of TH and D1-D2 modulators to ALBS modifications.

9. L.310: "critical" should be replaced with "important" or "significant".

10. L.401: Ref. 20, cited as evidence that HS is functionally defective in MPS, describes studies on MPS I, not MPS IIIA or MPS II. Since no ALBS are observed in MPS I patients, it is necessary to discuss that this reference is not directly connected to MPS types in which ALBS occur.

Reviewer #3 (Remarks to the Author):

The manuscript by De Risi et al. examines the behavioral, histological, and biochemical sequelae of

Sgsh deletion in rodent models of MPS-IIIa. They demonstrate a hyperdopaminergic phenotype in young mice that was autophagy-independent and caused by loss of heparin sulfate function. Additionally, they define the ontogeny of these phenotypes at multiple developmental time points, which is novel and much appreciated by this reviewer. I believe these findings will be of interest to the neurodevelopmental disorders and catecholamines research communities, as well as clinicians and health care providers working with MPS patients. Generally I found the manuscript to be thorough and well-written, although I have several questions/comments/concerns that should be considered and possibly addressed when preparing a revised manuscript. I have provided an itemized list below, which I hope the authors will find constructive.

1) I had difficulty examining the appropriateness of the statistical tests due to insufficient detail regarding methodology and data analysis in the Methods section. For example, it's unclear whether the authors used a within subjects or between subjects design, especially related to drug exposures. The methods state that they used a single group of 2-month old WT and MPS-IIIa mice (Page 14, Line 451); if true, did all mice receive all drug treatments? I think the answer is yes, as it says that mice were 're-injected in a counterbalanced order' on Page 16, Line 497; however, there are different numbers of mice in vehicle and drug groups, suggesting a between subjects design. Please clarify. Additionally, how many times was each behavior tested in each mouse? It would be helpful to provide more detail regarding the experimental protocol and timeline of drug exposure for each mouse, as polypharmacy and/or previous experience performing a task could potentially be confounds. A solution might be to create a schematic of the general experimental workflow that could be included in the supplement.

2) The methods appear to state that the Western blots and immunohistochemical analysis was performed after behavioral and drug testing was performed (Page 16, Line 498-99). Please provide justification that manipulating dopaminergic signaling with the pharmacological agents used does not alter D1R, D2R, or pDARPP-32 levels.

3) Given the morphological, electrophysiological, connectomic, and molecular differences between nigrostriatal and mesolimbic dopaminergic neurons, I have a philosophical objection to pooling VTA and SNc populations for analysis, especially when the justification is based on data that isn't shown (Page 16, Line 519). It is my opinion that the paper will be stronger if these populations are analyzed separately.

4) While the in vitro work examining mechanisms of dopaminergic neuron proliferation and phenotypic rescue with HS is strong, I'm less convinced based on the data that there is a D1-D2 imbalance. To clarify, I don't doubt the authors' interpretation of their findings, I just think the evidence that supports this hypothesis could be stronger. While D1R's are clearly post-synaptic on MSNs, there are both pre- and post-synaptic D2Rs in the striatum; could loss of D2 auto-inhibition "normalize" the total number of receptors while enhancing dopamine release? Additionally, the authors state that aripiprazole's partial agonist properties functionally balance the D1 and D2 tone, thus providing behavioral rescue. Aripiprazole produces a dose-dependent increase on ICSS thresholds in previous studies, suggesting it acts more like an antagonist than agonist on mesolimbic function. Additionally, the antagonist rescue studies were done at a single dose, which isn't that informative from a pharmacology perspective. If the authors have the time and/or expertise, I would suggest more thoroughly exploring D1 vs D2 contributions to phenotypic expression, even if it's doing something as simple as looking at the locomotor response to a range of D1 and D2R agonists and calculating the ED50 for each drug in wildtype and knockout mice.

5) Why are so many assays normalized to wildtype? For example, there's an internal control for the western blots, so why normalize normalized data to the wild type condition? For the dopaminergic neuron counts, why not report average counts per animal? Additionally, how is the 'wildtype' normalization performed? If the authors are calculating the mean of the wildtype group and normalizing all data points to that value, there should be little functional impact on the relative data variance between groups. Thus, nothing is being statistically gained and relevant experimental information (i.e. the actual magnitude of the raw values) is being lost.

Minor Issues:

1) Almost all statistical tests in the captions report a 1-way ANOVA, yet there are many cases where it appears that a 2-way ANOVA and/or repeated measures ANOVA was performed. Please check how all your statistical tests are reported to make sure they are correct.

2) The methods state that distance traveled, grooming, and rearing were all measured in the same session, suggesting that sample sizes should be identical for each metric. Unless I'm mistaken, in Figure 3, there appears to be some inconsistency in the number of data points within the same group from graph to graph (e.g. in vehicle treated knockout mice). Please check to make sure all data points are represented in the graphs; if any mice were omitted as outliers, please state so. It's possible that the points were automatically dropped by the graphing software due to 'data point crowding'; I know GraphPad Prism does this sometimes if the feature isn't turned off.

3) I may have missed it, but I don't believe 'TH' (tyrosine hydroxylase) was defined the first time it was used.

4) I find the sentence "Meso-striatal DA regulates almost all the behaviors that we found to be impaired..." (Page 6, Line 177) to be overly simplistic – there's a lot of subtlety in the role of dopaminergic signaling in diverse brain loci in the manifestation of these behaviors. It might be informative to provide more explanation of the scientific rationale of the subsequent experiments.

5) The authors refer to TH+ fibers in the SNr; please clarify whether these are SNc dopaminergic neuron dendritic arbors.

6) The number of statistics reported in each captions make them (the captions) difficult to read at times; perhaps a supplementary table with all statistical tests, sample sizes, and testing results would be both more informative and improve the readability of the captions, by allowing you to selectively report only the relevant statistics (unless this violates journal guidelines).

7) On Page 11, Line 359, change 'none' to 'no'.

8) On Page 12, Line 378, change 'receptors' to 'receptor'.

9) The methods section states that only male mice were used; although the authors provide an appropriate justification, it should clearly be stated in the introduction that this study only examined males to keep the implication of the results clear to readers, especially clinicians.

10) On Page 20, Line 633, there is an extra period at the end of the sentence.

11) In Figure 3a", the Y-axis goes to 120% of retreats. Would I be incorrect to assume that the maximum is 100%?

Point by point (in blue) response to the Reviewers for the manuscript: “Altered metabolism of heparan sulfate leads to developmental dopaminergic abnormalities responsible for autistic-like behaviours in lysosomal storage disorders”, NCOMMS-20-31163

Revision changes are highlighted in blue in the revised manuscript.

We thank the Reviewers for the high appreciation of the novelty, quality and relevance of our work. Despite the enormous difficulties due to the many lockdowns, we have done our best to address *with additional in vivo and in vitro experiments all the issues raised in the revision*. We are grateful to the Reviewers for helping in focusing on the additional pre-clinical evidence necessary to **support any therapeutic choice to manage autistic-like symptoms in MPS-III A** and to give broader relevance of the results in the context of the role of **HS as candidate player in autism-related neuropathological mechanisms**. We sincerely think that the revised version is improved and more straightforward and hope that it is ready for publication in *Nature Communications*.

REVIEWER COMMENTS

Reviewer #1 (Remarks to the Author):

In this study, De Risi et al have explored a potential novel cause for the autistic-like behavioral changes seen in the lysosomal storage disease: mucopolysaccharidosis type III (Sanfilippo syndrome). Using a previously well validated mouse model with loss of function of the Sgsh gene, they find that these behaviors are related to hyperdopaminergia, in turn apparently due to developmental increases in dopamine producing cells of the midbrain, and can be ameliorated with drugs that affect the dopamine pathway. These behaviors form a very significant source of morbidity and distress for individuals with MPSIII A and their carers, so identifying a new affected pathway and potential for treatment would be of significant clinical benefit, and the results of this study will be of interest to the broad community. The authors have used a wide range of approaches to investigate their hypothesis which overall I find broadly convincing. I think the findings of this study have the potential to be exciting, however I have a number of comments on the manuscript.

Major comments

A key argument made in this study is that there is an increase in the D1R pathway, with unaffected D2Rs. The authors use either a D1R antagonist or a D2R agonist –Aripiprazole –to rescue some of the autistic-like behavioral changes seen in the Sgsh-/- mouse, either way ‘rebalancing’ the D1/D2 activity. My issue here is that Aripiprazole is a D2R partial agonist –and is often thought to decrease D2 activity (at least in the mesolimbic pathway) as it is less active than dopamine itself. While it may well be more complex than this, I think for me this section would be much more compelling if the authors could show that, for example, classic D2R antagonists such as haloperidol or risperidone do NOT have the same beneficial effects.

We thank the Reviewer for appreciation of our work and for carefully helping us to improve the manuscript.

We agree with the Reviewer that a direct comparison between D1 and D2 antagonist in this model would provide further support to the evidenced D1/D2 imbalance and provide an important pre-clinical evidence. In the revised manuscript, we have included a treatment with haloperidol as suggested by the Reviewer (Fig. 2f-f' and Fig. 3c-c'', attached below). As expected, based on previous studies, haloperidol increased P-DARPP-32 in the striatum of WT animals (Valjent et al., Neuropsychopharmacology 2011) but not in MPS-IIIa mice, likely due to a ceiling effect (Fig. 2f-f'). It did not improve sociability and stereotypies at either of the two doses used. In the open field test, the highest dose of haloperidol (0.1 mg/kg) decreased locomotor activity in MPS-IIIa mice as compared to vehicle treatment (Fig. 3c-c''). Therefore, if anything, haloperidol has only minor effects on the hyperactivity phenotype. We have reported these novel data in the revised manuscript (pages 7, lines 196-198; page 7, lines 214-219). Challenged by a request from Reviewer 3 on this same issue, we have found that D2Rs pre-synaptic expression is decreased in MPS-IIIa striatum as compared to WT littermates. By contrast, D2Rs post-synaptic expression is increased, although this increase is much lower as compared to that observed for D1Rs (15% vs 49%, respectively). Thus, reduced D2 pre-synaptic inhibitory action further supports hyperdopaminergia in MPS-IIIa mice, which results in overstimulation of the D1-direct pathway, as evidenced by increased DARPP-32 at the post-synaptic level. This provides further support on the mechanisms limiting the efficacy of haloperidol in MPS-IIIa (page 12, line 372-373).

Figure 2f-f'. MPS-IIIa showed hyperactivation of DARPP-32 intracellular signalling, which is rescued by SCH-23390 treatment but not by haloperidol. Total level of DARPP-32 was not changed. Representative WB for each condition is presented. * $p < 0.05$ vs WT, between groups; # $p < 0.05$ vs MPS-IIIa veh, within group.

Figure 3c-c'. Haloperidol (0.05 and 0.1 mg/kg) rescued hyperactivity, only at higher doses, while had no effect on stereotyped behaviours and social interaction impairment. Histograms represent mean \pm S.E.M. * $p < 0.05$ vs WT, between groups; # $p < 0.05$ vs MPS-IIIa veh, within group.

Concerning Aripiprazole, previous findings have shown that Aripiprazole acts as an agonist and antagonist depending on the dopaminergic status (De Bartolomeis et al., CNS Drugs 2015). However, we agree with the Reviewer that any interpretation is an oversimplification. In the current manuscript we have eight main and 12 supplementary figures. Unless we consider splitting it into two manuscripts, to publish in a back-to-back story,

adding Aripiprazole would swell the manuscript excessively and deviate too much from the novel important evidence on the neurodevelopmental dopaminergic defect in two of the lysosomal storage disorders. Therefore, we think it would be better to publish it in a separate manuscript.

Related to this, the authors used α -MPT to rescue some behaviors, but this drug is known to cause hypoactivity at lower doses than used here (Khakimova et al, Mol Neurobiol 2017). Can the authors include its effect on the social interaction test – this would be the better test.

The Reviewer is right, Khakimova et al, Mol Neurobiol 2017 reported that α -MPT induces hypoactivity 4 hours after the administration of 170 mg/kg. This is line with our data showing that 250 mg/kg of α -MPT leads to severe hypoactivity in both genotypes five hours after the injection (Supplementary Fig. 3g). Previous literature showed that α -MPT-mediated effects are also time-dependent (Kesner et al., Psychopharmacology 1977; Watanabe et al., Journal of Oral Science 2005; French et al., Biochemical Pharmacology 2005). Indeed, we tested mice 2 hours after the injection. Consistent with the time-course of motor effects, although a moderate (50%) reduction of DA release in the striatum can be observed 2 hrs post-injection, the depletion is still not complete, like 5 hr after (Watanabe et al., Journal of Oral Science 2005).

Concerning the social interaction tests, we decided to use only the social tube test for ethical reasons. α -MPT is harmful for the animals and it has long-lasting effects. Therefore, as we have explained better in the revised version of the manuscript (pages 16, lines 489-501), we tested the mice in the social tube test immediately followed by the open field test on the same day and avoided the re-injection of α -MPT on the following days. We selected the social tube test because it has a lower mean duration time with respect to the social novelty task, thus avoiding a longer delay from injection. Adding further groups of animals treated with α -MPT would not have been approved by the ethical committee for animal use in experimentation, due to the lack of sufficient translational relevance and novelty as compared to evidence we have already collected.

Another really important part of this manuscript is the quantification of increased numbers of dopaminergic cells (using TH staining), and at several developmental stages. However, the methodology used could be improved and the details given are insufficient. Overall, given the different approaches (including HPLC, Western blots, etc.) I am convinced that the key finding of increased activity in this pathway is probably true, but they could be more compelling. It would be preferable to use fluorescence immunolabelling and double label with at the very least DAPI, and ideally a neuronal marker like NeuN.

It would be nice to see a comparison with other cell types in the same region (as opposed to hippocampus NeuN which is unrelated) and to confirm whether overall cell numbers are changed.

We thank the Reviewer for addressing this important point. As requested by the Reviewer, we have added to the immunohistochemical count an immunofluorescence analysis for TH, NeuN and Parvalbumin (PV), on WT and MPS-IIIa mice at 2-months-old, in order to validate the same finding (TH+ cell increase) and to evaluate if the cell numbers of another neuronal population changed. As expected, MPS-IIIa mice showed increased numbers of TH+ neurons in both the SN and the VTA, compared to WT (Supplementary Fig. 4a-b, attached below). In contrast, PV+ neurons were not significantly increased in SN and VTA of MPS-IIIa mice (Supplementary Fig. 4b'). Interestingly, considering the overall number of NeuN+ cells, it was increased in MPS-IIIa with respect to WT. We reasoned that this increase was due to the cells being double-stained with TH and NeuN. Indeed, considering only the number of NeuN+ that were not labeled with TH (NeuN+/TH-),

there was no significant difference between WT and MPS-IIIa mice, in both SN and VTA (Supplementary Fig. 4c-c'). The number of DAPI+ nuclei never changed (Supplementary Fig. 4d). We have reported these new data in the revised version of the manuscript (pages 8, lines 230-235) and removed the hippocampal experiment so as not to make the manuscript too long. We thank the Reviewer for this comment, which further strengthens our conclusions.

Supplementary Figure 4. (a-b) Immunofluorescence analysis confirmed the increased number of TH+ cells in SN and VTA of MPS-IIIa mice, compared to WT. (b'-d) There was no difference, instead, in the number of PV+ neurons or in the number of DAPI nuclei between WT and MPS-IIIa. The number of total NeuN+ neurons was also increased. However, when considering only the number of NeuN+ neurons, not colocalized with TH+, there was no difference between WT and MPS-IIIa. A representative staining for each condition is presented. Histograms represent mean \pm S.E.M.

As all data are presented as normalized values, it is not clear whether these are numbers of cells or densities of cells (both would be good), nor whether they come from regions of interest (ROIs) or the entire region. If the

former, how was ROI location chosen? If the latter, how was this delineated? Additional details on quantification are needed –were experiments performed and analyzed blind, were data from sections averaged? The description of the densitometry is unclear and needs more detail (would not be able to reproduce), and it is not clear to me how this accounts for variability in DAB staining per se.

We apologize for the lack of clarity of the methodology section. We have now included additional details (pages 18-19, lines 581-607) and references to previous works using identical procedures (Luk et al., Science 2012; Giordano et al., Brain 2018), as follow:

The number of TH+ cells were manually counted on 20x magnification images using ImageJ software (NIH; Bethesda, MD), by an experimenter blind to the genotype. For each experimental group, regularly spaced 50 μ m sections were chosen at multiple rostrocaudal levels in order to sample the SN and VTA. Specifically, we sampled considering the following anteroposterior coordinates relative to the bregma (according to Franklin & Paxinos, 2001): -2.92, -3.16; -3.28; -3.52; -3.64. Each experimental group section (WT or MPS-IIIa and WT or MPS-II) was matched for each coordinate. Cell numbers from each section were averaged and expressed as percentage from WT, as previously done (Luk et al Science 2012; Giordano et al Brain 2018). The identification and the definition of subregions (SN and VTA) were done based on the mouse brain atlas (Franklin & Paxinos, 2001) and on cytoarchitectonic characteristics of the two areas (i.e. cell orientation) (Krashia et al., European Journal of Neuroscience 2017; Fu et al., Brain Struct Funct 2012). We delineated the area representative of SN and VTA in the main figures. We used the same methodology for immunofluorescence analysis on TH, NeuN and PV neurons. For p0, regularly spaced 20 μ m sections were similarly chosen. We counted within the entire area delimited by DAB staining.

Similarly, for E13.5 mice, we performed cell counting as described above, by counting all slices, regularly spaced 20 μ m, between the coronal section 18 and the coronal section 21 slices following the prenatal mouse brain atlas Uta Schambra (2008). For these slices, we counted within the entire area delimited by DAB staining. We used the same methodology for BrdU and LMX1A counting.

We have evaluated TH fiber immunoreactivity by measuring the optical density in three squared boxes, positioned at different points of the SN. Quantification was done on three regularly spaced 50 μ m bilateral sections (corresponding approximately to bregma -3.08; -3.52; -3.64). Optical density values were corrected by subtracting OD from a non-stained area of the same slice and expressed as percentage from WT from the same immunostaining batch to correct for differences in DAB immunoreactivity (Decressac et al., Neurobiology of Disease 2012).

The in vitro cell culture section nicely complements the in vivo work, and overall is convincing. However, I am surprised by the finding of increased TH cell number at 4DIV but not at 7DIV. This appears to not make sense, as if there was higher proliferation early on, there should still be more cells 3 days later. Is there any evidence of cells dying or of lysosomal degradation at this stage (given that MAP2 seems unchanged, no generalized apoptosis)?

Also, they should quantify the total number of cells (eg using DAPI) and give density values, as density effects can be pronounced in culture.

Can the authors comment on why there is such a discrepancy with in vivo timing of proliferation and cell death?

We thank the Reviewer for this comment because we realized that expressing the *in vitro* data in terms of percentage to WT generated confusion. As suggested by the Reviewer, in the revised version of the manuscript, we reported the density values [(TH+ cells/DAPI)*100] clearly showing that there is a time-dependent increase in the number of TH+ cells over the days in WT cells (Fig. 5b-c, attached below). In contrast, in MPS-III A cells we report a higher TH+ cell density at DIV4, no differences at DIV7 and a significant decrease at DIV14 (Fig. 5b-c.). A similar pattern was seen for MAP2+ cells [(MAP2+ cells/DAPI)*100] (Fig. 5b-c).

Figure 5b. Primary cultures of mesencephalic neurons reproduce *in vitro* the progression of the TH phenotype, showing an increase in dopaminergic cells density at DIV4. However, while in WT cells there was a time-dependent increase in TH+ neurons, MPS-III A cells showed a decrease of TH+ cell density at DIV14. **Figure 5c.** MAP2+ cell density neurons was not changed at DIV4 and DIV7, while it was reduced at DIV14 in MPS-III A cells. Histograms represent mean \pm S.E.M. * $p < 0.05$ vs WT, between groups; # $p < 0.05$ vs WT, within groups; $^{\circ}p < 0.05$ vs MPS-III A, within groups.

This pattern of a change in TH+ cell density *in vitro* recapitulates the different stages observed *in vivo*, as defined by pre- and after-lysosomal dysfunction. We added immunofluorescence experiments on autophagy/lysosomal markers (p62, LAMP-1 and TUNEL staining), showing that although LAMP-1+ spots increased already at DIV4 (similar to what happens *in vivo*), p62+ spots and TUNEL density were increased only at DIV14, indicative of an autophagy/lysosomal dysfunction associated with generalized apoptosis at this late stage (Supplementary Figure 6, attached).

Supplementary Figure 6. Immunofluorescence analysis revealed that there was an increase in the number of p62+ and LAMP-1+ spots in MPS-III A primary neurons at DIV14 but not at DIV4. This impairment was associated to an increased density of TUNEL+ neurons. A representative staining for each condition is presented. Histograms represent mean \pm S.E.M. * $p < 0.05$ vs WT, between groups.

We obtained similar results for iDA (Supplementary Figure 7b-c, attached below). Due to a lack of figure space, in the revised manuscript we have moved these experiments from the main figure (Fig. 6) to the supplementary figure 7.

Supplementary Fig. 7b-c. Induced dopaminergic neurons (iDA) from MPS-III A embryonic fibroblasts (MEF) showed a higher percentage of TH+ cells at DIV4 and a progressive loss at DIV14. Tuj1+ neurons were increased at DIV4, while were reduced at DIV14 in MPS-III A cells. Histograms represent mean \pm S.E.M. * $p < 0.05$ vs WT, between groups; # $p < 0.05$ vs MPS-III A veh, within group.

Minor comments

In the introduction (line 94) the authors say ‘[haloperidol 90 and risperidone] ... have considerably higher than expected extrapyramidal side effects in these patients’ but ref 7 shows risperidone is effective with no extrapyramidal side effects. While I appreciate this sentence is ambiguous, I think the authors should be more circumspect in their claim here.

We thank the Reviewer for noticing the incongruence, the higher extrapyramidal side effects were reported by ref 6 (Tchan et al., J Intellect Dev Disabil 2009), which is also included in line 90. The Reviewer is correct that the literature gives conflicting results. For instance, extrapyramidal effects after risperidone treatment were also reported in a child affected by MPS-IIIB (Santosh et al., BMJ Case Rep. 2009). A deeper analysis of the literature suggests that Risperidone is mainly effective on hyperactivity and the extrapyramidal effects are mainly reported in older patients, which is in line with the side effects reported after these antipsychotic drugs in neurodegenerative disorders. We modified the sentence in the revised version of the manuscript trying to take into account the conflicting literature on the use of antipsychotics in MPS, as follow: “they have little or variable therapeutic effects on autistic-like behaviours in MPS-IIIA, mainly on hyperactivity in very young children; however, they have been reported to have considerably higher than expected extrapyramidal side effects in older patients” (page 3, lines 92-95).

Behavior –

No direct test of olfaction was done. This either needs to be added, or if this has previously been done (in their hands preferably) please comment and cite.

We agree with the Reviewer that testing olfaction is important to rule out an important intervenient variable in the social interaction test. We have included a further group of animals tested in an odour task with a habituation/dishabituation paradigm identical to the social task. The data show that two-month-old MPS-IIIA mice have no odour discrimination deficits, which rules out its possible contribution in the lack of interest for the conspecific (Supplementary Fig. 1, attached below). The detailed procedure was described in supplementary methods.

*Supplementary Fig. 1a. No difference was evident between WT and MPS-IIIA 2-month-old mice, in the odour discrimination test. Histograms represent mean ± S.E.M. *p<0.05 new vs familiar, within group.*

Some behaviors in the methods lack a statement as to whether done / scored blind.

We thank the Reviewer for noticing. We have added a sentence stating that all our experiments are done by experimenters' blind to the genotype and the treatment (page 15, lines 471-472).

Can the authors comment on the repetitive behaviors they see? –these are not a feature of individuals with MPSIIIA, nor are seen in the previous published study of the Sgsh-/- mouse.

Repetitive behaviours were not systematically tested in MPS-IIIa children (Rumsey et al., J Pediatr 2014). However, parents of children affected by MPS-IIIa through the Repetitive Behaviour Questionnaire-2 (RBQ-2), that evaluates the presence of autistic-like repetitive behaviours and restricted rituals and routines (such as repetitive motor stereotypies, sensory interests, rituals and routines), evidenced common social and communication difficulties in their children. 53% of children were also reported to “Almost always choose from a restricted range of repetitive activities” and “Play the same music, game or video, or read the same book repeatedly” (Wolfenden et al., Br J Learn Disabil 2018), which is line with the high degree of misdiagnosis of autism often occurring with MPS-III children (Wijburg et al., Acta Paediatrica 2013). We have detailed the description of these behaviours in the revised manuscript (page 11; lines 352-354).

Aripiprazole is already licenced for treatment of irritability in autism, but can have many side-effects and sometimes exacerbation of symptoms. While this does not invalidate the findings here (this might be a good example of how to determine who would benefit), the manuscript would benefit from a discussion of this.

After the revision, we have added many experiments, including haloperidol treatment, which further supports the D1-D2 hypothesis. We agree with the Reviewer that the discussion on the clinical relevance of Aripiprazole for MPS-IIIa needs to be deepened. Based on the mechanistic evidence reported in this manuscript we are planning to perform acute and chronic experiments to test far more treatments for autistic-like symptoms in MPS-IIIa. We think it is better to include the results of Aripiprazole in an upcoming manuscript comparing all these treatments to provide clinicians with pre-clinical evidence to support their therapeutic choice for MPS-IIIa. We hope that the Reviewer agrees with our decision.

This is entirely personal preference, but I think inventing unnecessary new acronyms is best avoided. ‘ALBs’ is not a standard acronym and not one I have ever encountered, and I would prefer not to use it. Also – I think it is unhelpful, as it is better to refer to the specific behavior.

We have removed “ALBSs” from the whole manuscript.

Social dominance – please give more details. 10cm is very short (eg the reference given uses 30cm). Why are all the percentages 33 66 100 for WT and different for mutant? Also – please call ‘social interaction’ social dominance.

We apologize for the error. The apparatus used for the social tube test was 30 cm length as reported in Irie et al., PNAS 2012. The percentages 33, 66 and 100 correspond to 1, 2 and 3 retreats occurring in the three-trials test, respectively, according to the formula: [(number of exits or retreats/total number of trials)*100]. We explained this point better in the revised version of the manuscript (page 17, lines 531-534).

In the figure, when we wrote “social interaction”, we referred to both the social tube and the social novelty task. In the revised version of the manuscript, we were more specific and used “social dominance” for the social tube and “social interaction” for the social novelty task (Fig. 1; Fig.3; Fig. 8; Supplementary Fig. 2).

Social novelty – at 8 months, seems that the WT aren’t interested in stranger mouse, as opposed to any change in the mutants.

We thank the Reviewer for noticing this peculiar result. In 8-month-old WT mice, there was a single mouse that had an extreme value (0% in the sociability index, black arrow in the figure appended below), that we did not exclude in the previous version of the manuscript.

In the current version of the manuscript, we added a further small group of WT and MPS-III A at 8-months (N=3 WT and N=2 MPS-III A). Hence, we performed the outliers test (Z-score function) and the extreme mouse was excluded from all the analysis (as reported in the revised version of the manuscript, Fig. 1a-e). No differences are evident between WT at 2 and 8 months of age. This information is now reported in the revised version of the manuscript (page 15, lines 485-487).

Dopamine/TH analysis –

Line 187 – state age (is on figure, but would be good in text too).

We added this information (page 6, line 173).

Please give more details of D1R analysis in methods – just says used Metamorph.

We apologize for the lack of detailed description. To address an important point raised by the Reviewer 3, which is in line with many of the comments made by the Reviewer 1, we have performed a further analysis to quantify pre- and post-synaptic D2R expression in the striatum and included a detailed description of the methods in the revised version of the manuscript (pages 20; lines 618-625) as follows:

“The quantitative analysis of the relative abundance of D1R or D2R and colocalization between D2R with synaptophysin and/or PSD95 was performed per μm^2 of coronal sections of Caudate (DMS), Putamen (DLS) and shell of accumbens nucleus (nAcc) brain areas of WT or MPS-III A mice using a Leica Metamorph imaging software (Leica Meta-Morph, Germany) according to the method reported in Laperchia et al., Brain Struct. Funct 2017. In brief, the distribution density of D1R or D2R/synaptophysin/PSD-95 multiple labelled puncta was counted in a region of interest of $4 \times 10^4 \mu\text{m}^2$ for DMS or DLS or nAcc brain area through $5 \mu\text{m}$ depth thickness by considering $n = 200$ serial Z-stacks ($0.5 \mu\text{m}$ each) per region. $N = 10$ serial brain sections/mouse”.

Why pool data from SN and VTA? Please outline key regions (eg SN, VTA) on images (and to indicate exactly where quantifications done from)

In the current version of the manuscript, we show separate graphs for SN and VTA and delineated the area in the figures (Fig. 4a-b; Fig. 8g-g', Supplementary Fig. 4; Supplementary Fig. 5a-b).

Line 233-6 – why state that this suggests an accelerated cell loss? Surely going from more cells to normal is expected in this context?

Line 239-40 - ‘At 2-months of age, no cells positive for cc3+ (Fig. S2f), suggesting that behavioral symptoms (ALBSs and DLSs) and the TH phenotype progressively worsen in parallel with the lysosomal/autophagosomal dysfunction and HS accumulation’ –not clear to me what is being said here. Surely, there is progressive loss of many cell types, so the hyperdopaminergia is gradually lost (indeed, initially returned to normal levels?), while the loss of other neurons leads to dementia?

We thank the Reviewer for this comment. Actually, we have also found cc3+ cells in the hippocampus of 8 month-old MPS-IIIa mice, but we did not report this in the revised manuscript due to a lack of space. We have removed the sentence in the revised manuscript.

Fig 4c (BrdU) –looks like picked non-representative spot for high power view, at least for anterior mesencephalon.

We modified the spot.

Fig 7 – no scale bars (and are MPSIII cells as same magnification as WT in DIV4 images? (Nuclei size looks different).

Thank you for noticing it, we added the scale bar. The Reviewer is right: nuclei look larger, but we did not address this aspect in manuscript. Therefore, in the revised version we have mentioned it, without speculating on the functional meaning (page 10, line 296).

Line 313-5 – I think it is a stretch to say this suggests that there is a loss of HS function in the brain in MPSIIIa (nor other MPS disorders), although this would be very interesting to test directly, if possible. If not, a more nuanced phrasing would be more appropriate.

Fgf2 experiments – this needs a bit more explanation – please briefly justify rationale for this experiment in the results section as well as discussion, please include in methods (currently completely missing) and state what Baf32 cells are.

We apologize for missing information.

The Baf32 cells are lymphoblastoid cells that overexpress the FGFR1 and they proliferate when HS and FGF2 are added to the medium, as HS acts as co-receptor of FGF2 when acting on FGFR1; therefore, mitogenicity of Baf32 cells is generally used to test the functional interaction between HS and FGF2. We thank the Reviewer for this comment and we have added this information in the revised version of the manuscript (pages 10-11, lines 324-328; supplementary methods page 8).

As reported in Supplementary Fig. 11, addition of HS extracted from WT, but not from MPS-IIIa, brains leads to proliferation. Based on this evidence we concluded that there is an altered HS function in MPS-IIIa brain; we explained better this experiment in the revised version of the manuscript, and we nuanced the conclusion referring to “altered HS function” (page 2, line 41; page 9, line 266; page 10, lines 307, 315, 317, 322; page 11, line 330-331, 334; page 13, line 422).

Please include full Western blots in the supplementary information.

We included source data showing uncropped blots.

It is very difficult to see scatter plot over dark blue bars, please use different colour/line combination.

We changed the colour of data points (white instead of the colour of bars).

Minor corrections

Fig 2A – what is the picture of? Not mentioned in legend.

Line 77 – ‘incapacitating’ instead of ‘invalidating’?

Line 96 – reference 8 does not look at dopaminergic cells – only colocalization of NCAM with HS in layer 5 cortex.

Lin 107 – neurexin

Line 183 – GABA’ergic’

Line 503 – one half, not each half

Line 239 – no cells *were* positive...

Fig 4b – missing scale bars for P0 and P28

Line 269 – ‘autophagosomal’

Fig 5c legend ‘ MAP2+ neurons were *not*

Line 313 – actually Fig S6 not S5

Line 350 – ‘which is rescued’ – need to be clear that have only rescued in vitro effects, not those in embryo.

We thank the Reviewer for helping us in improving the quality of our manuscript, we performed all changes as suggested.

Reviewer #2 (Remarks to the Author):

The question asked in this work - what is the molecular mechanism of autistic-like symptoms in mucopolysaccharidosis (MPS) type IIIA - is excellent and important to understand the disease. On the basis of a series of well-conducted experiments, the authors demonstrated developmental dopaminergic abnormalities in the MPS IIIA mouse model which were partially confirmed in the MPS II mouse model. Experiments with cellular models of MPS IIIA suggested that there are changes in heparan sulfate (not only accumulation of this glycosaminoglycan) which may contribute to observed biochemical abnormalities. This report can be a significant contribution to the field, indicating previously unknown molecular mechanisms of the disease. Nevertheless, I have several comments which I suggest should be addressed.

We sincerely thank the Reviewer for carefully reviewing, for positively evaluating our manuscript and for helping to address important points related to the different MPS subtypes and the link with the structural changes in HS composition in MPS-III.

Major points:

1. The title suggests that "altered metabolism of heparan sulfate (HS) leads to developmental dopaminergic abnormalities responsible for autistic-like symptoms in lysosomal storage disorders". However, only MPS IIIA and MPS II models were studied. Therefore, I suggest to restrict the conclusions, as well as the title, to these two diseases.

We restricted the conclusion to MPS-II and MPS-III (page 2, lines 45-47; pages 14, lines 453-454). For the title, however, we would prefer not to specify the two pathologies, as rare diseases such as MPS-II and III are not known among scientists working in neurodegenerative disorders and/or neurodevelopmental disorders. By contrast, lysosomal dysfunction is a hot topic in these fields and by referring to LSDs, which not necessarily mean "all LSDs", we hope to make the manuscript more attractive for people working in tangential fields.

2. The problem of too general conclusions regarding the effects of HS on dopaminergic abnormalities and resultant autistic-like symptoms (see point 1) arises from the fact that no (or only little) autistic-like symptoms are observed in patients suffering from other diseases in which HS accumulation (and changed metabolism) is evident, like in MPS I. This should be discussed in the light of the proposed mechanism.

We thank the Reviewer for challenging this issue, as it allows us to better contextualize our results. The Reviewer is correct that autistic-like behaviours in MPS-I children has attracted less attention than in MPS-III. However, it has been recently reported that also MPS-I children show high scores in Social Responsiveness Scale (SRS) (suggesting social impairments), associated with other neuropsychiatric alterations such as depression, attention problems, aggressive behaviour and sleep disorders (Lehtonen et al., JIMD Reports 2017). In line with findings in MPS-II, autistic-like symptoms in MPS-I might be masked by a much more severe and faster phenotype (for review see Barone et al., Ital J Pediatr 2014), which we show in this manuscript progresses in parallel with the neurodevelopmental defects. We have addressed this point in the revised manuscript (page 3, lines 81-85; page 14, lines 445-448).

3. Undegraded (or rather partially degraded) HS is stored in MPS IIIA cells. However, addition of normal HS or HS derived from WT organisms partially restored selected defects (Fig. 8 b and b'). From the discussion it appears that the authors suggest that HS from WT and MPS organisms differs considerably. Therefore, chemical structures of HS derived from WT and MPS IIIA animals should be determined to indicate if there are significant chemical changes between these molecules which might be responsible for their functional modulations.

The Reviewer is correct that the partial rescue in *in vitro* experiments with commercial and HS derived from WT brains suggests that the chemical structure of HS in MPS-III A is altered. We have realized that, in the previous version of the manuscript, we had not discussed our results in the light of previous evidence in the literature that already analysed the changes in HS composition and sulfation in MPS-III A mouse brains and provided evidence that this is affected at different levels, including the degree of sulfation and accumulation of non-reducing end (NRE) (Maccari et al., *Metab. Brain Dis.* 2017; Dwyer et al., *Sci. Rep.* 2017). In particular, disaccharide analysis showed an overall increase in the charge density, with a clear increase in the degree of 2-O-sulfation in MPS-III A (already in very young mice) (Maccari et al., *Metab. Brain Dis.* 2017; Dwyer et al., *Sci. Rep.* 2017). The partial rescue in MPS-III A SH-SY5Y cells by WT HS suggested that it might compete with the endogenous hypersulfated HS for the binding of growth factors and partially normalize proliferation. Other structural changes previously reported in MPS-III A (Maccari et al., *Metab. Brain Dis.* 2017; Dwyer et al., *Sci. Rep.* 2017) and MPS-I mouse models (Holley et al., *JBC* 2001), such as the degree of sulfation, the presence of more NRE, the core protein structures, the localization of HS (cell membrane vs. intracellular) could modify the outcome of the binding between HS-growth factor-receptor (Gambarini et al., *Molecular and Cellular Biochemistry* 1993; Schlessinger et al., *Molecular Cell* 2000; Wu et al., *J. Biol. Chem* 2003; Zhang et al., *J. Biol. Chem*). Therefore, we decided to directly probe whether HS from MPS-III A brain shows altered binding capacity for FGF2, using the BaF32 assay. The BaF32 assay is a lymphoblastoid cell line that lacks cell surface HS and overexpresses the FGF receptor type 1 (FGFR1), which proliferates in response to FGF2 only if functional HS is added to cell culture medium. Hence, we compared the capacity of HS extracted from WT and MPS-III A mouse brains to induce FGF2-FGFR1-dependent mitogenicity. The BaF32 assay confirmed that HS from MPS-III A had an altered function as a co-receptor, in line with similar findings reported in cells from MPS-IIIB patients (De Pasquale et al., *Molecular Therapy* MCD 2018) or in other MPS with altered HS metabolism, reporting altered HS-FGF2 signalling (Pan et al., *Hematopoiesis* 2005).

In our future studies we will address the highly interesting issue addressing all the structural, expression and localization changes occurring in HS and related HSPG in MPS-III A and their functional neuronal consequences in the different stages of the disease. We thank the Reviewer for the suggestion, and we have addressed this issue in the discussion in the revised manuscript (page 14, lines 424-433).

4. Although behavioral tests used in this work are generally accepted as reflecting ALBS in mice, it is necessary to indicate that a mouse model was used in which autism might be, nevertheless, quite different than in humans. It is true that there are no better models, thus, the experimental system is fine, but the limitations in interpretation of the results should be discussed.

We agree with the Reviewer and we realized that we referred to “autistic-like symptoms” in mice might be misleading; we have changed it with “autistic-like behaviours”. Additionally, we discussed this point in the current version of the manuscript (pages 12; lines 359-365) and reported below:

“Although the behavioural tests used in this study are aimed at dissecting species-specific behaviours and therefore they can phenotypically differ from the clinical manifestation in humans, they are generally accepted as reflecting autistic-like and dementia-like symptoms in mice. Additionally, they allowed to separate early occurring and late occurring behavioural manifestations that are associated with the absence and presence, respectively, of autophagosomal dysfunctions, therefore providing evidence that the MPS-IIIa animal model recapitulates the progression of the pathology.”

5. Excess of normal HS did not completely rescue the measured parameters shown in Fig. 8 (panels b and b') but rather improved them towards values estimated for WT controls. This should be indicated in the text, and interpreted not as correction (rescue) but as improvement. As a consequence, HS dysfunction should not be presented as the cause of the observed biochemical and behavioral changes, but rather its contribution to the pathomechanism can be suggested.

We thank the Reviewer for this important comment and we have substituted the word “rescued” with “reduced” and we have nuanced the conclusion, discussing the possible mechanisms (page 2, line 41; page 9, line 266; page 10, lines 307, 315, 317, 322; page 11, line 330-331, 334; page 13, line 422).

Minor points:

6. L.107-108: Neurexin and neuroligin are not genes. There are proteins, encoded by corresponding genes. The authors should clearly distinguish whether they refer to genes or proteins throughout the text.

We apologize for these errors. We corrected them in the revised version of the manuscript.

7. L.110-127: Description of the results of this study is too long in Introduction. At the end of this chapter, rationale of the study should be presented rather than quite long description of obtained results and conclusions.

We thank the Reviewer for this suggestion, we have shortened the description of the results and evidenced the rationale of the study as follow (page 4; line 112-117):

“Using the Sgsh^{-/-} mouse model (hereafter called MPS-IIIa), bearing a spontaneous mutation in the sulfamidase gene, we have investigated the changes in the development and function of the mesostriatal dopaminergic system in MPS-IIIa and their sensitivity to dopaminergic drugs. Using different in vitro models of MPS-IIIa (primary mesencephalic neurons, induced dopaminergic neurons from embryonic fibroblasts and neuroblastoma cell line knocked-out for the gene SGSH by CRISPR/Cas9), we tested the role of HS in regulating the dopaminergic dysfunction in MPS-IIIa.”

8. L.222: "indicating a first causal disease mechanism" is a too strong statement. ALBS is a complex syndrome, thus, the interpretation of the results can be presented as either suggestion or possible contribution of TH and D1-D2 modulators to ALBS modifications.

We agree with the Reviewer and we have modified the text as follow: “These data provide proof-of-concept that preventing TH activation or blocking D1R activation rescue hyperactivity and social interaction deficits

in MPS-IIIa mice, indicating a possible disease mechanism and disease specific therapeutic approaches for autistic-like behaviour in MPS-IIIa” (page 7, lines 219-222).

9. L.310: "critical" should be replaced with "important" or "significant".

We replaced it with “significant”.

10. L.401: Ref. 20, cited as evidence that HS is functionally defective in MPS, describes studies on MPS I, not MPS IIIa or MPS II. Since no ALBS are observed in MPS I patients, it is necessary to discuss that this reference is not directly connected to MPS types in which ALBS occur.

The Reviewer is aware that MPS-I can present with a large spectrum of clinical severity. As in MPS-II pathology the accumulations of other GAGs favors the progression of the lysosomal dysfunction. Therefore, the early onset of dementia can easily mask autistic-like symptoms in these pathologies. However, recent clinical evidence in milder forms of MPS-I reports the presence of an autistic-like symptoms also in MPS-I. We thank the Reviewer for challenging this issue, as it allows to discuss in the manuscript this important point as follow:

“They also manifest in other forms of mucopolysaccharidosis (MPS) characterized by defective metabolism of HS, such as MPS-II and MPS-I; although, in these pathologies, they can be more difficult to detect as the concomitant defective metabolism of the enzyme iduronate-2-sulfatase and α L-iduronidase, respectively, leads to an accumulation of HS and other GAGs, resulting in a faster shift from autistic- to dementia-like symptoms” (page 3, lines 81-85).

“All these findings are also relevant for MPS-II and even if our data are restricted to these two LSDs, they might be also relevant for other LSDs showing altered metabolism of HS, such as MPS-I, where autistic-like behaviours, including social impairment, attention problems and aggressive behaviour have been described” (page 14; lines 445-448).

Reviewer #3 (Remarks to the Author):

The manuscript by De Risi et al. examines the behavioral, histological, and biochemical sequelae of Sgsh deletion in rodent models of MPS-III A. They demonstrate a hyperdopaminergic phenotype in young mice that was autophagy-independent and caused by loss of heparin sulfate function. Additionally, they define the ontogeny of these phenotypes at multiple developmental time points, which is novel and much appreciated by this reviewer. I believe these findings will be of interest to the neurodevelopmental disorders and catecholamines research communities, as well as clinicians and health care providers working with MPS patients. Generally I found the manuscript to be thorough and well-written, although I have several questions/comments/concerns that should be considered and possibly addressed when preparing a revised manuscript. I have provided an itemized list below, which I hope the authors will find constructive.

We thank the Reviewer for evaluating our manuscript, for highlighting the relevance of our work in the field of neurodevelopmental disorders and for the useful comments, which helped us to further strength our conclusions.

- 1) I had difficulty examining the appropriateness of the statistical tests due to insufficient detail regarding methodology and data analysis in the Methods section. For example, it’s unclear whether the authors used a within subjects or between subjects design, especially related to drug exposures. The methods state that they used a single group of 2-month old WT and MPS-III A mice (Page 14, Line 451); if true, did all mice receive all drug treatments? I think the answer is yes, as it says that mice were ‘re-injected in a counterbalanced order’ on Page 16, Line 497; however, there are different numbers of mice in vehicle and drug groups, suggesting a between subjects design. Please clarify.

We apologize for the lack of details. We used a Latin square design and all animals received all drug treatments for SCH and haloperidol experiments. As opposite, we used a between-subjects design for α -MPT experiment.

In the previous version of the manuscript, the number of the animals was not the same for all experimental groups because: (i) we included in the statistics the animals that were tested in the open field test and used for western blot on DARPP-32, which we did not submit to the other behavioural tests; (ii) some animals were not tested in the social tasks due to the lack of availability of a conspecific or of a WT mouse on the specific testing-day. In the current version of the manuscript, not to generate confusion, we removed from the statistics the mice used for the biochemical analysis and those that did not perform all the tests. In the revised version, according to the request of the Reviewer, we have added another cohort of animals tested with a Latin square design to test the 0.06 mg/kg dose of SCH 23390, resulting in the total number of animals per group reported in the table below and in the revised manuscript (page 16, lines 489-518).

Experiment I (SCH 0.03)	Experiment II (SCH 0.06)	Total
WT veh = 6	WT veh =8	WT veh = 14
WT SCH 0.03=6	WT SCH 0.06=8	WT SCH 0.03=6; WT SCH 0.06=8
MPS-III A veh=8	MPS-III A veh=7	MPS-III A veh = 15
MPS-III A SCH 0.03=8	MPS-III A SCH 0.06=7	MPS-III A SCH 0.03=8; MPS-III A SCH 0.06=7

We apologize for not reporting this information in the previous version of the manuscript.

Additionally, how many times was each behaviour tested in each mouse? It would be helpful to provide more detail regarding the experimental protocol and timeline of drug exposure for each mouse, as polypharmacy

and/or previous experience performing a task could potentially be confounds. A solution might be to create a schematic of the general experimental workflow that could be included in the supplement.

We thank the Reviewer for this comment, and we have included a detailed description of the experimental design in the method section, by describing each experiment separately (page 16, lines 489-518). Those of SCH-23390 have been tested twice as we originally planned only one dose, while those treated with haloperidol were tested 3 times because we had 2 doses. We also added a schema of the experimental design in the supplementary methods (attached below), as suggested by the Reviewer.

(A) Cohort 1 – SCH-23390 treatment

(B) Cohort 2 – SCH-23390 treatment

(C) Cohort 3 – Haloperidol treatment

2) The methods appear to state that the Western blots and immunohistochemical analysis was performed after behavioral and drug testing was performed (Page 16, Line 498-99). Please provide justification that manipulating dopaminergic signaling with the pharmacological agents used does not alter D1R, D2R, or pDARPP-32 levels.

We again apologize for the missing information. The biochemical and histological analysis on D1R, D2R, TH etc. were performed on drug-naïve mice, as described in the revised version of the manuscript (page 15, lines 475-487; page 16, lines 519-523), and reported below:

“Experiment 1. Behavioural, biochemical and histological characterization of MPS-IIIa and MPS-II. We tested MPS-IIIa, MPS-II and their WT littermate mice at different time points (MPS-IIIa: at 2, 6 and 8 months of age and MPS-II: at 1-month-old of age) in a battery of behavioural tasks (see the experimental scheme in supplementary methods) to assess autistic-like and dementia-like behaviours (related to Fig. 1, Fig. 8, Supplementary Fig. 2). At the end of the behavioural tasks, mice were used for biochemical (TH, dopamine receptors, DA content etc.) and histological characterization (related to Fig. 2b-e', Fig. 4a-a', Fig. 8f-g', Supplementary Fig. 3a-f, Supplementary Fig. 4, Supplementary Fig. 5, Supplementary Fig. 12)”.

Regarding the experiments on pDARPP-32 levels, we used a further group of mice (not submitted to Latin square design, as described below). We reported below the information included in the revised version of the manuscript:

“Experiment 4. Effects of SCH-23390 and haloperidol on DARPP32 pathway. To perform the biochemical analysis on DARPP32 pathway (related to Fig. 2f-f’), a group of mice was injected with the drug or vehicle, submitted to the open field test and then sacrificed by cervical dislocation. Striatal tissues were used for western blot. Two mice injected with SCH-23390 were excluded from the statistical analysis, as they resulted outliers, according to Z-score function”.

3) Given the morphological, electrophysiological, connectomic, and molecular differences between nigrostriatal and mesolimbic dopaminergic neurons, I have a philosophical objection to pooling VTA and SNc populations for analysis, especially when the justification is based on data that isn’t shown (Page 16, Line 519). It is my opinion that the paper will be stronger if these populations are analyzed separately.

We agree with the Reviewer and in the revised manuscript we have reported separately the VTA and the SN (Fig. 4a-b; Fig. 8g; Supplementary Fig. 4; Supplementary Fig. 5a-b), even if the data indicate a significant effect of the variable Genotype, but not of the variable Brain region.

4) While the in vitro work examining mechanisms of dopaminergic neuron proliferation and phenotypic rescue with HS is strong, I’m less convinced based on the data that there is a D1-D2 imbalance. To clarify, I don’t doubt the authors’ interpretation of their findings, I just think the evidence that supports this hypothesis could be stronger. While D1R’s are clearly post-synaptic on MSNs, there are both pre- and post-synaptic D2Rs in the striatum; could loss of D2 auto-inhibition “normalize” the total number of receptors while enhancing dopamine release? Additionally, the authors state that aripiprazole’s partial agonist properties functionally balance the D1 and D2 tone, thus providing behavioral rescue. Aripiprazole produces a dose-dependent increase on ICSS thresholds in previous studies, suggesting it acts more like an antagonist than agonist on mesolimbic function. Additionally, the antagonist rescue studies were done at a single dose, which isn’t that informative from a pharmacology perspective. If the authors have the time and/or expertise, I would suggest more thoroughly exploring D1 vs D2 contributions to phenotypic expression, even if it’s doing something as simple as looking at the locomotor response to a range of D1 and D2R agonists and calculating the ED50 for each drug in wildtype and knockout mice.

We are grateful to the Reviewer of this comment, which challenged our biochemical data suggesting that the lack of increase in D2Rs expression could be masked by a differential change in the pre- and post-synaptic components. Therefore, we addressed this issue by performing a triple immunostaining for D2 with a pre- and post-synaptic marker, synaptophysin and post-synaptic density protein 95 (PSD-95), respectively (Fig. 2e’, attached below). In line with the prevision of the Reviewer, we have found that D2Rs pre-synaptic expression is decreased in MPS-IIIa striatum as compared to WT littermates. By contrast, D2Rs post-synaptic expression is increased, although this increase is much lower as compared to that observed for D1Rs (15% vs 49%, respectively). Thus, reduced D2 pre-synaptic inhibitory action further supports hyperdopaminergia in MPS-IIIa mice, which results in overstimulation of the D1-direct pathway, as evidenced by increased DARPP-32 at the post-synaptic level.

Fig. 2e'. Triple immunolabeling against D2R, SYN and PSD-95 revealed a decreased co-localization of D2R/SYN and an increased colocalization of D2R/PSD-95, suggesting a decreased and increased expression of D2R at presynaptic and postsynaptic level respectively. Representative images of D2R/SYN/PSD-95 immunofluorescence of WT and MPS-III A mice are shown. Enlargement of the boxed area showed D2R/SYN coexpression as yellow puncta (arrows) and D2R/PSD-95 as violet puncta (arrows). Scale bar: 50 μ m. Histograms represent mean \pm S.E.M. * $p < 0.05$ vs WT, between groups.

In line with these findings and with the request of the Reviewers (including Reviewer 1), we have added another dose of SCH-23390 (Fig. 3b-b", attached below), which confirmed that normalizing DARPP-32 expression is associated with the rescue of all the behavioural defects observed in MPS-III A mice. We also added haloperidol, at two doses that were effective in increasing DARPP-32 signalling in WT animals, while leading to blunted effects in MPS-III A mice. The latter did not rescue the behavioural phenotype (Fig. 3c-c", attached below). We think that all together these results support our general hypothesis that hyperdopaminergia in MPS-III A mice leads to behavioural symptoms through an upregulation of the D1-direct pathway, which is normalized by D1R antagonists.

Fig. 3b-b''; Fig. 3c-c''. SCH-23390 (0.03 and 0.06 mg/kg) rescued hyperactivity, stereotypies and social impairment. Haloperidol (0.05 and 0.1 mg/kg) rescued hyperactivity, only at higher doses, while had no effect on the stereotypies and social impairment. Histograms represent mean \pm S.E.M. * p <0.05 vs WT, between groups; # p <0.05 vs MPS-III A veh, within group.

As explained to the Reviewer 1, we agree that the data concerning Aripiprazole need further discussion that for the lack of space we cannot do in this manuscript, therefore we have removed it from the results. We thank the Reviewer for his/her comment, and we hope that he/she will be satisfied, as we are, with the way we have addressed his/her concerns.

5) Why are so many assays normalized to wildtype? For example, there's an internal control for the western blots, so why normalize normalized data to the wild type condition? For the dopaminergic neuron counts, why not report average counts per animal? Additionally, how is the 'wildtype' normalization performed? If the authors are calculating the mean of the wildtype group and normalizing all data points to that value, there should be little functional impact on the relative data variance between groups. Thus, nothing is being statistically gained and relevant experimental information (i.e. the actual magnitude of the raw values) is being lost.

For western blot experiments, we reported normalized data to the WT group because we calculated the percentage relative to the control for each blot, to correct for any type of differences relative to different gels (for example recycling of primary antibody or time to ECL exposition). Similarly, for IHC experiments we calculated the percentage relative to the control from the same immunostaining batch to correct for differences

in DAB immunoreactivity. This information has now been reported in the revised version of the manuscript (page 20, lines 637-639). Where it was possible, we have reported absolute values (ie. TH counting in *in vitro* experiments).

Minor Issues:

1) Almost all statistical tests in the captions report a 1-way ANOVA, yet there are many cases where it appears that a 2-way ANOVA and/or repeated measures ANOVA was performed. Please check how all your statistical tests are reported to make sure they are correct.

The Reviewer is correct, we corrected the description of the analysis.

2) The methods state that distance traveled, grooming, and rearing were all measured in the same session, suggesting that sample sizes should be identical for each metric. Unless I'm mistaken, in Figure 3, there appears to be some inconsistency in the number of data points within the same group from graph to graph (e.g. in vehicle treated knockout mice). Please check to make sure all data points are represented in the graphs; if any mice were omitted as outliers, please state so. It's possible that the points were automatically dropped by the graphing software due to 'data point crowding'; I know GraphPad Prism does this sometimes if the feature isn't turned off.

We realized that some data points were overlapped. We thank the Reviewer for this suggestion and we now turned off this function on GraphPad.

3) I may have missed it, but I don't believe 'TH' (tyrosine hydroxylase) was defined the first time it was used.

We corrected it (page 6, line 172).

4) I find the sentence "Meso-striatal DA regulates almost all the behaviors that we found to be impaired..." (Page 6, Line 177) to be overly simplistic – there's a lot of subtlety in the role of dopaminergic signaling in diverse brain loci in the manifestation of these behaviors. It might be informative to provide more explanation of the scientific rationale of the subsequent experiments.

The idea of testing the functionality of the dopaminergic system in MPS-IIIa originated from the long-standing experience of our lab on the physiological role of dopamine in the accumbens core and shell and in the dorsomedial-lateral striatum, in controlling hyperactivity, social interaction and stereotyped behaviours (De Leonibus et al., Eur. J. Neurosci. 2006; Manago et al., Learn Mem. 2009; De Leonibus et al., Psychopharmacology 2007), and on the clinical evidence that these children might present higher than expected extrapyramidal side effects in response to dopaminergic drugs. We are aware that striatal subcircuits have a different role in regulating each of these behaviours. However, in MPS-IIIa we have found that all of them were altered, thus suggesting a hyperactivation of the whole circuit.

In the revised manuscript, we have changed the sentence as follows: 'Hyperactivity, social interaction and stereotyped behaviours are abnormal behaviours subtly regulated by ventral to dorsal meso-striatal and meso-

cortical circuits (De Leonibus et al., Eur. J. Neurosci. 2006; Bariselli et al., Nature communications 2018). To understand the neurobiological mechanisms leading to autistic-like behaviours in MPS-IIIa and to design specific therapeutic approaches, we focused our biochemical analysis on DA at the level of the corpus striatum (including both the ventral and the dorsal parts), which is one of the main terminal regions of mesencephalic DA neurons” (page 6, lines 162-167).

5) The authors refer to TH+ fibers in the SNr; please clarify whether these are SNc dopaminergic neuron dendritic arbors.

The Reviewer is right. We incorrectly wrote TH+ “fibers”, instead that “dendritic arbours”. The quantification was done in SNr but refers to the cells in SNc. To avoid confusion, we used “SN” in the revised version of the manuscript (page 19, lines 601-607).

6) The number of statistics reported in each captions make them (the captions) difficult to read at times; perhaps a supplementary table with all statistical tests, sample sizes, and testing results would be both more informative and improve the readability of the captions, by allowing you to selectively report only the relevant statistics (unless this violates journal guidelines).

We added supplementary tables as suggested by the Reviewer (Supplementary Table 2 and 3).

7) On Page 11, Line 359, change ‘none’ to ‘no’.

8) On Page 12, Line 378, change ‘receptors’ to ‘receptor’.

We corrected all these errors.

9) The methods section states that only male mice were used; although the authors provide an appropriate justification, it should clearly be stated in the introduction that this study only examined males to keep the implication of the results clear to readers, especially clinicians.

We added a sentence in the first paragraph of the results to better clarify this point, as suggested by the Reviewer (page 4, lines 125-126).

10) On Page 20, Line 633, there is an extra period at the end of the sentence.

11) In Figure 3a”, the Y-axis goes to 120% of retreats. Would I be incorrect to assume that the maximum is 100%?

We corrected all these errors.

Reviewer #1 (Remarks to the Author):

The authors have now carried out a significant number of additional experiments. Most importantly from my perspective, they have examined the effect of haloperidol (a traditional D2 antagonist) and present compelling evidence that this does not have the same beneficial effects on behavior as the D1 antagonist. They have also decided to drop their previous data on aripiprazole. I personally agree with this decision – I think the data they present now is clear, and for the reasons I originally gave, it is hard to interpret the aripiprazole experiments. They have addressed all my concerns very comprehensively, including further additional experiments to significantly improve the quantification and assessment of increased dopaminergic cells, as well as providing all the detail necessary in the methods. I have no further concerns.

I also provide a (non-comprehensive) list of proof-reading suggestions:

Line 205 – ‘social interaction’ to ‘social dominance’

Figure 4 – would be nice to label SN and VTA on image.

Line 268 – associated with...

Line 273 – ‘As opposite’ – ‘conversely..’?

Line 278 – there was no difference...

Line 294 – ‘confirmed that our’

Fig 6 – seem to have lost DIV labels on several graphs.

Line 352 – not ‘which show’... these are individuals – ‘who...’

Reviewer #2 (Remarks to the Author):

The authors have addressed all my points indicated in the previous round of review. Thus, I found the revised manuscript acceptable for publication.

Grzegorz Wegrzyn

Reviewer #3 (Remarks to the Author):

The authors have sufficiently addressed all concerns I had with the previous version of the manuscript. Overall, I believe the authors' extra work revising and adding additional data to the manuscript has paid off by improving the interpretability and impact of the findings. I look forward to seeing this paper in press. Kudos.

Point by point (in blue) response to the Reviewers for the manuscript: “Altered heparan sulfate metabolism during development triggers dopamine-dependent autistic-behaviours in models of lysosomal storage disorders”, NCOMMS-20-31163

REVIEWER COMMENTS

Reviewer #1 (Remarks to the Author):

The authors have now carried out a significant number of additional experiments. Most importantly from my perspective, they have examined the effect of haloperidol (a traditional D2 antagonist) and present compelling evidence that this does not have the same beneficial effects on behavior as the D1 antagonist. They have also decided to drop their previous data on aripiprazole. I personally agree with this decision – I think the data they present now is clear, and for the reasons I originally gave, it is hard to interpret the aripiprazole experiments. They have addressed all my concerns very comprehensively, including further additional experiments to significantly improve the quantification and assessment of increased dopaminergic cells, as well as providing all the detail necessary in the methods. I have no further concerns.

I also provide a (non-comprehensive) list of proof-reading suggestions:

Line 205 – ‘social interaction’ to ‘social dominance’

Figure 4 – would be nice to label SN and VTA on image.

Line 268 – associated with...

Line 273 – ‘As opposite’ – ‘conversely..’?

Line 278 – there was no difference...

Line 294 – ‘confirmed that our’

Fig 6 – seem to have lost DIV labels on several graphs.

Line 352 – not ‘which show’ ... these are individuals – ‘who...’

We thank the Reviewer for all corrections, we have made them in the revised manuscript (highlighted in blue).